# *Supervise Less, See More*: Training-free Nuclear Instance Segmentation with Prototype-Guided Prompting

Wen Zhang [* 1 2]   Qin Ren [* 1]   Wenjing Liu [1]   Haibin Ling [1]   Chenyu You [1]

## Abstract

Accurate nuclear instance segmentation is a pivotal task in computational pathology, supporting data-driven clinical insights and facilitating downstream translational applications. While large vision foundation models have shown promise for zero-shot biomedical segmentation, most existing approaches still depend on dense supervision and computationally expensive fine-tuning. Consequently, training-free methods present a compelling research direction, yet remain largely unexplored. In this work, we introduce **SPROUT**, a fully training- and annotation-free prompting framework for nuclear instance segmentation. SPROUT leverages histology-informed priors to construct slide-specific reference prototypes that mitigate domain gaps. These prototypes progressively guide feature alignment through a partial optimal transport scheme. The resulting foreground and background features are transformed into positive and negative point prompts, enabling the Segment Anything Model (SAM) to produce precise nuclear delineations without any parameter updates. Extensive experiments across multiple histopathology benchmarks demonstrate that SPROUT achieves competitive performance without supervision or retraining, establishing a novel paradigm for scalable, training-free nuclear instance segmentation in pathology. Code is available at https://github.com/Y-Research-SBU/SPROUT.

## 1. Introduction

Nuclear instance segmentation delineates individual nuclei for systematic downstream analysis (Caicedo et al., 2019; Greenwald et al., 2022; Gupta et al., 2023) and advances cancer prognosis, diagnosis, and treatment (Madabhushi & Lee, 2016; Lu et al., 2018; Pinckaers et al., 2021). In histopathology, cellular structure visualizations are most commonly obtained from hematoxylin and eosin (H&E) staining (Vahadane et al., 2016). Hematoxylin highlights nuclei in dark blue or purple and eosin stains cytoplasm in pink for separation (Ruifrok et al., 2001). However, the intrinsic properties of pathology images pose unique challenges for instance segmentation. First, the narrow color spectrum and staining variability limit robust visual cues. Second, a single patch can contain thousands of densely packed nuclei with weak boundaries. Third, pixel-wise annotations are scarce and labor-intensive for pathologists.

To address these challenges, numerous specialized nuclear segmentation networks have been explored under varying levels of supervision, including fully-supervised (Graham et al., 2019; Qu et al., 2019; He et al., 2021b; Chen et al., 2023a; He et al., 2023), semi-supervised (Zhou et al., 2020; Wu et al., 2022; Jin et al., 2022), weakly-supervised (Zhao & Yin, 2020; Nishimura et al., 2021; Liu et al., 2022), and self-supervised (Sahasrabudhe et al., 2020; Xie et al., 2020). Nevertheless, their performance is often limited under distribution shifts, varying annotation protocols, and restricted training data (Pachitariu & Stringer, 2022). Together, these limitations underscore the pressing need for generalizable and robust approaches to nuclear instance segmentation.

Vision foundation models mark a turning point in image segmentation. Segment Anything Model (SAM) (Kirillov et al., 2023) achieves zero-shot, class-agnostic segmentation by leveraging large-scale training on the SA-1B datasets. Building on SAM, subsequent work has explored fine-tuning (Zhang et al., 2024c; Ma et al., 2024; Peng et al., 2024; Archit et al., 2025) and adapter-based strategies (Chen et al., 2023b; Na et al., 2024; Cheng et al., 2024; Chen et al., 2025a) to accommodate SAM to medical objectives. However, the need for substantial annotations and resource-intensive training constrains their practicality in pathology.

An appealing alternative is to transform feature correspondences between the annotated references and targeted images into actionable prompts (Liu et al., 2024; Zhang et al., 2024a; Liu et al., 2025) without supervision or retrain-

---

[*]Equal contribution [1]Stony Brook University, NY, USA [2]Johns Hopkins University, MD, USA. Correspondence to: Chenyu You <chenyu.you@stonybrook.edu>.

*Proceedings of the 43rd International Conference on Machine Learning*, Seoul, South Korea. PMLR 306, 2026. Copyright 2026 by the author(s).

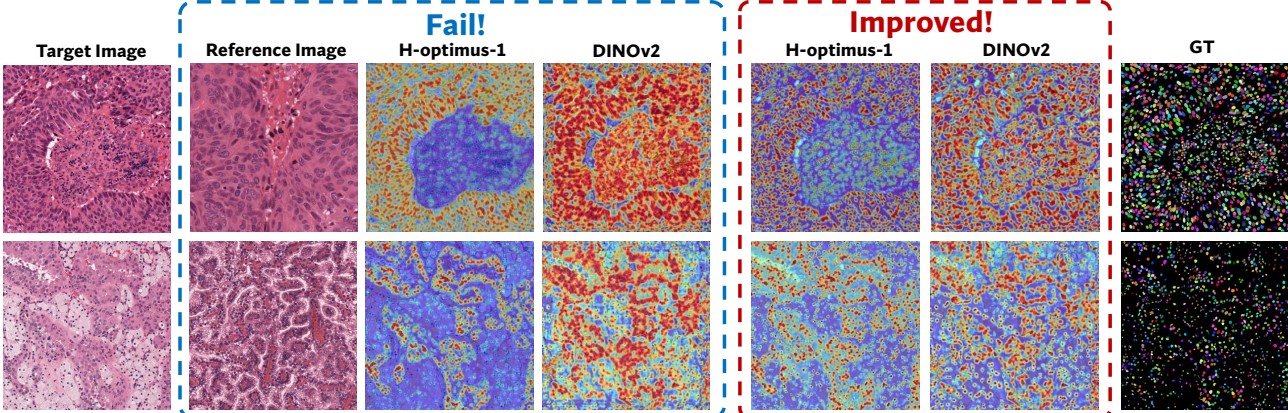

*Figure 1.* **Comparison of one-shot and proposed self-reference strategies in feature extraction.** One-shot (**blue box**) fails to capture precise and diverse nuclei, even with similar pairs or backbones trained on natural and pathology images. Instead, our self-reference approach (**red box**) leverages high-confidence regions within the image to extract more robust features for similarity guidance.

ing. One might reasonably expect obtaining high-quality prompts from external backbones trained on natural or pathological domains, such as DINOv2 (Oquab et al., 2024) and H-optimus-1 (Bioptimus, 2025). Yet the distinct properties of pathological images hinder such direct transfer. As illustrated in Figure 1, even when target and reference pairs are carefully matched in color, nuclear size, and spatial distribution, backbone models exhibit characteristic failures. Feature extractors trained on natural images can over-amplify the nuclear regions, while pathology-trained ones still struggle to capture subtle, heterogeneous cell-level features. Unlike natural images with a limited number of salient objects occupying large portions, nuclei are harder to capture reliably due to their fine-grained structures, which may consist of only a few thousand pixels. Moreover, few-shot strategies are often impractical in pathology. Variations in staining, cellular density, and local morphology preclude the establishment of appropriate references or consistent image matching. Consequently, their unstable performance across references is unsuitable for practical deployment.

In this work, we address these challenges by introducing **SPROUT** (**S**tain **P**riors with p**R**ototypical partial **O**ptimal transport for **U**nlabeled promp**T**ing), a novel prompting framework that guides SAM in nuclear instance segmentation *without any training*. Specifically, we leverage the biochemical affinity of H&E staining to generate high-confidence foreground and background regions, from which semantically reliable prototypes are extracted to inform global feature allocation. Such a *self-reference* strategy is notably effective as these prototypes calibrate image-specific feature activations for precise and complete nuclear delineation, as illustrated in Figure 1. To achieve comprehensive coverage of the cells and mitigate ambiguity, we propose **POT-Scan**, a principled progressive partial optimal transport scheme built on feature–prototype similarity mapping. In addition, we incorporate biological priors into

a containment-aware Non-Maximum Suppression (NMS) strategy for SAM prediction refinement. We empirically validate our SPROUT across four benchmark datasets, *i.e.*, MoNuSeg (Kumar et al., 2017), CPM17 (Vu et al., 2019), TNBC (Naylor et al., 2018), and PanNuke (Gamper et al., 2020), and demonstrate remarkable performance compared with SAM-based, fully-supervised, and weakly-supervised counterparts while remaining computationally efficient. Our main contributions are summarized as follows:

- To the best of our knowledge, SPROUT is the first fully training-free framework for nuclear instance segmentation in H&E pathology images without annotations. We introduce a novel, lightweight, and generalizable self-reference mechanism that overcomes reference-based limitations and bridges domain gaps.

- We propose POT-Scan, a principled scheme with theoretical guarantees that adaptively balances nuclear coverage and noise suppression. Our quantitative and qualitative analyses further elucidate the intrinsic behavior of prompt generation and verify its robust performance under diverse hyperparameter settings.

- Across four challenging benchmarks, SPROUT consistently achieves remarkable performance gains (+8.2% AJI on MoNuSeg). These results highlight the potential of robust prompt generation and patch-based decomposition to unlock the zero-shot capabilities of vision foundation models in histopathology.

## 2. Related Work

**Prompt Engineering for SAM.** SAM (Kirillov et al., 2023), as a vision foundation model, enables zero-shot and class-agnostic segmentation via point, box, and coarse mask prompts. However, its performance in precise object delineation heavily depends on prompt accuracy and placement. This has spurred a growing body of work on auto-

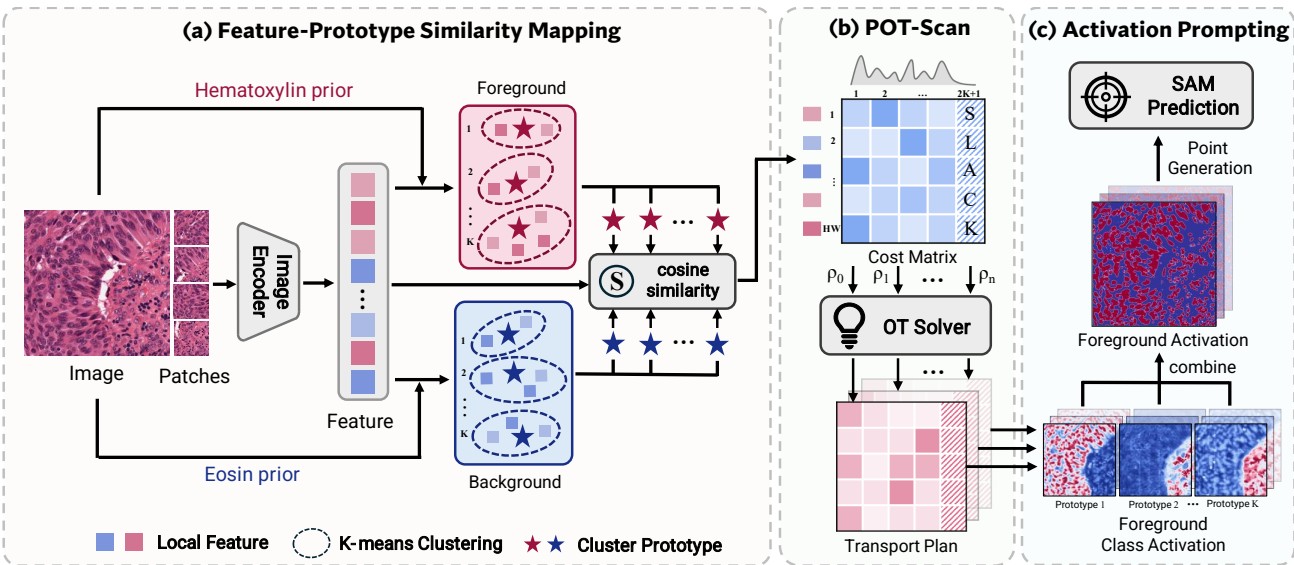

*Figure 2.* **SPROUT pipeline for point prompt generation.** It consists of three steps: (i) Feature–prototype similarity mapping: H&E stain priors identify high-confidence foreground and background regions, from which clustering extracts representative prototypes that serve as anchors for similarity matching; (ii) POT-Scan: a partial optimal transport scheme that progressively aligns features to prototypes, filtering ambiguous assignments through partial mass transport; (iii) Activation prompting: prototype-reweighted activations are aggregated into foreground maps, from which positive and negative point prompts are sampled to guide SAM-based instance prediction.

matic prompt generation, as large-scale manual guidance is impractical in clinical settings. Recent work includes embedding-oriented prompt representation learning (Luo et al., 2023; Yue et al., 2024; Li et al., 2025; Yan et al., 2025), detector-based prompt generation (Wu et al., 2023; Zheng et al., 2024; Xu et al., 2024; Xie et al., 2025), and heuristic-driven approaches (Gao et al., 2024). Meanwhile, prototype-guided methods (Zhang et al., 2024c; Wang et al., 2025) exploit feature correspondences between reference and target images to improve representation learning. Building on this idea, several training-free approaches generate prompts directly from such similarity mapping in natural images (Zhang et al., 2024a; Liu et al., 2024) and medical images (Liu et al., 2025). However, these methods are typically designed for few-object scenarios and rely on carefully curated reference images. By contrast, nuclear segmentation poses a more challenging problem, involving thousands of densely packed, morphologically diverse objects within a single image. These limitations highlight the need for new strategies for reliable prompt generation without external references while remaining computationally efficient.

**Nuclear Instance Segmentation.** Fully supervised nuclear instance segmentation approaches can be broadly grouped into three categories: contour-based (Chen et al., 2016), distance-mapping (Graham et al., 2019; He et al., 2021a), and detection-based (Jiang et al., 2023). While effective, these methods heavily rely on dense pixel-level annotations, which are costly and time-consuming. Semi- and weakly-supervised alternatives, particularly point-supervised approaches, have been explored as a new direction by trans-

forming sparse labels into coarse pixel-level cues, such as Voronoi-based (Tian et al., 2020) or pseudo-edge maps (Yoo et al., 2019). But these approaches still suffer from unreliable pseudo masks and fail to adequately separate overlapping nuclei. With the advent of SAM, adapting foundation models to medical imaging has become a prominent direction. Methods such as MedSAM (Ma et al., 2024) and fine-tuning variants (Huang et al., 2024a;b) still require additional annotations and incur substantial computational overhead. To alleviate these constraints, other studies, including PromptNucSeg (Shui et al., 2024), UN-SAM (Chen et al., 2025b), and All-in-SAM (Cui et al., 2024), attempt to reduce these costs by training auxiliary prompters, introducing domain-adaptive feature tokens, or fine-tuning from self-generated masks. Building on these, we argue that competitive segmentation performance can be attained directly from SAM through proper prompt design and appropriate patch granularity without altering the model architecture.

## 3. SPROUT

The pipeline comprises three steps: (i) feature–prototype similarity mapping (Section 3.1), (ii) partial optimal transport scan with activation prompting (Section 3.2), and (iii) instance mask prediction with refinement (Section 3.3). As shown in Figure 2, SPROUT leverages stain priors to construct robust self-reference features, aligns them with semantically reliable prototypes through a theoretically grounded POT-Scan, and generates precise point prompts to guide SAM without additional training. The following subsections describe each stage in detail. The detailed theoretical

analysis is provided in Appendix A.

### 3.1. Feature-prototype Similarity Mapping

The first step is to extract self-reference features from stain priors and condense them into representative prototypes for subsequent matching. Given an image $I \in \mathbb{R}^{H \times W \times 3}$, we partition it into $n$ overlapped patches $\{I^i\}_{i=1}^n$. Each patch is encoded by a pretrained encoder $f_\theta$ into $F^i = f_\theta(I^i)$. Patch features are stitched to reconstruct a global representation $F \in \mathbb{R}^{h \times w \times d}$, where $(h, w)$ is the spatial resolution of the encoded feature map and $d$ is the embedding dimension.

To derive self-reference masks, we condition on stain color priors by transforming images into optical density space: $OD = -\log(x/x_0)$, where $x$ is the observed RGB intensity and $x_0$ is the reference intensity. Using the normalized stain matrix $Q = [Q_H, Q_E]$, the concentration map $S = [S_H, S_E]^\top$ is obtained via linear decomposition: $S = Q^+ \cdot OD$, where $Q^+$ is the pseudoinverse. See Appendix B.5 for the pipeline schematic. Otsu's thresholding (Appendix B.8) is then utilized to separate coarse foreground and background regions. Within each region, pixels with the top $t$ stain intensities are selected to construct high-confidence masks $M_{fg}$ and $M_{bg}$. To ensure spatial correspondence, we resize the pixel-level masks to feature resolution and overlap them for prototype extraction. Each class-specific feature set is clustered into $K$ groups using $K$-means to derive representative prototypes $\mathcal{P}_c = \{P_c^1, \ldots, P_c^K\}$:

$$\mathcal{P}_c = \underset{\{P_c^1, \ldots, P_c^K\}}{\arg\min} \sum_{p \in \Omega} M_c(p) \min_{k=1,\ldots,K} ||F(p) - P_c^k||^2, \quad (1)$$

where $c \in \{fg, bg\}$, $M_c \in \{0, 1\}$, and $\Omega$ denotes the feature map spatial locations. The resulting prototypes $\{\mathcal{P}_{fg}, \mathcal{P}_{bg}\}$ capture the global semantic feature characteristics of foreground and background, serving as class-specific anchors for subsequent mapping.

### 3.2. POT-Scan and Activation Prompting

How can we reliably propagate prototype semantics to all features when isolated point-wise assignments are unstable and collapse-prone in the presence of noise? Given the intra-class variability of foreground and background features, a distribution-level formulation that allocates features coherently across prototypes is needed.

**Preliminary.** Optimal transport (OT) seeks the minimal-cost mapping between distributions under marginal constraints, providing a principled measure of distributional discrepancy. Given probability vectors $\mu \in \mathbb{R}^{n \times 1}$, $\nu \in \mathbb{R}^{m \times 1}$ with cost matrix $C \in \mathbb{R}_+^{n \times m}$, the Kantorovich formulation (Kantorovich, 1942) solves:

$$\min_{T \in \mathbb{R}^{n \times m}} \langle T, C \rangle_F, \quad \text{s.t. } T\mathbf{1}_m = \mu, \ T^\top \mathbf{1}_n = \nu, \quad (2)$$

where $T$ is the transport plan and $\langle \cdot, \cdot \rangle_F$ denotes the Frobenius inner product. Relaxing the marginal constraints with divergences yields unbalanced OT. Adding entropic regularization (Cuturi, 2013) enables efficient solutions via the Sinkhorn–Knopp algorithm (Knight, 2008). Further background and solver details are in Appendix A.1 and A.2.

**POT-Scan.** While OT offers a principled formulation, assigning features to prototypes in pathology is challenging due to noise and ambiguity. A naïve approach is to match features to their nearest prototypes via cosine similarity:

$$C = 1 - \frac{\tilde{F}P^\top}{||\tilde{F}||_2 ||P||_2}, \quad (3)$$

where $P \in \mathbb{R}^{2K \times d}$ is the prototype matrix and $\tilde{F} \in \mathbb{R}^{hw \times d}$ is the flattened feature map. However, standard OT requires full mass transport, and even its unbalanced variant still imposes penalties for discarding noisy features. To address this, we adopt **partial OT**, which allows a fraction of the mass to remain unmatched and naturally filters ambiguous regions (Appendix A.3). Formally, transporting a fraction $\rho \in (0, 1]$ of the mass is posed as:

$$\min_{T \in \Pi} \ \langle T, C \rangle_F + \lambda KL(T^\top \mathbf{1}_N || \frac{\rho}{M} \mathbf{1}_M),$$
$$\text{s.t. } \Pi = \{T \in \mathbb{R}_+^{N \times M} | T\mathbf{1}_M \le \frac{1}{N}\mathbf{1}_N, \ \mathbf{1}_M T^\top \mathbf{1}_N = \rho\} \quad (4)$$

where $N = h \times w$ denotes the number of source features, assumed to follow a uniform mass distribution since each feature is equally weighted, and $M = 2K$ is the number of target prototypes. This leads to POT-Scan, where the transport ratio $\rho$ is progressively increased: starting from a small initial value $\rho_0$ that favors easy feature–prototype matches and gradually incorporating more ambiguous features until a stopping criterion is met.

Although conceptually intuitive, Eq.(4) cannot be solved directly using standard scaling algorithms because of non-normalized constraints. Following the reformulation of (Zhang et al., 2024b), we append a slack column to absorb the residual $1 - \rho$ mass, thereby restoring normalized marginals and enabling efficient Sinkhorn-based optimization. Detailed proofs, solver derivation, and corresponding pseudo-code are provided in Appendix A.3, A.4, and A.5.

**Activation Prompting.** Given optimal transport plan $T^\star$, features are reweighted as $F^\star = \tilde{F} \odot T^\star \in \mathbb{R}^{hw \times 2K}$. These are mapped back to the image space via the resizing operator $\mathcal{R}$ and refined with DenseCRF, producing activation maps $F' = \text{CRF}(\mathcal{R}(F^\star)) \in \mathbb{R}^{H \times W \times 2K}$. Foreground and background activations are aggregated as $[F'_{fg}, F'_{bg}] = [\sum_k F'^k_{fg}, \sum_k F'^k_{bg}]$, which are then binarized using Otsu's thresholding. Combining these with the initial high-confidence masks yields positive points through a watershed-based procedure (Appendix B.8), while negative

*Table 1.* **Performance evaluations of nuclear instance segmentation.** We benchmark methods across fully-, weakly-, and self-supervised, and SAM-based approaches on MoNuSeg and CPM17 datasets. Segmentation performance is reported using AJI (↑), PQ (↑), DQ (↑), SQ (↑), and Dice (↑). Best results are highlighted in **bold**, and second-best are underlined.

| Method | SAM | Supervision | MoNuSeg | | | | | CPM17 | | | | |
|---|---|---|---|---|---|---|---|---|---|---|---|---|
| | | | **AJI** | **PQ** | **DQ** | **SQ** | **Dice** | **AJI** | **PQ** | **DQ** | **SQ** | **Dice** |
| U-NetMICCAI'15 | ✗ | fully | 0.421 | 0.403 | 0.571 | 0.705 | 0.635 | 0.554 | 0.527 | 0.718 | 0.734 | 0.741 |
| SPN+IENISBI'22 | ✗ | point | 0.521 | 0.436 | 0.661 | 0.660 | 0.677 | 0.540 | 0.485 | 0.695 | 0.699 | 0.701 |
| SC-NetMEDIA'23 | ✗ | point | 0.539 | 0.450 | 0.648 | 0.694 | 0.732 | 0.561 | 0.486 | 0.692 | 0.703 | 0.698 |
| SAMCVPR'23 | ✓ | point* | 0.061 | 0.262 | 0.384 | 0.751 | 0.353 | 0.135 | 0.469 | 0.601 | 0.781 | 0.329 |
| DES-SAMMICCAI'24 | ✓ | box | 0.463 | 0.429 | 0.621 | 0.691 | 0.672 | 0.512 | 0.517 | 0.735 | 0.704 | 0.688 |
| MedSAMNat. Commun.'24 | ✓ | box* | 0.502 | 0.327 | 0.514 | **0.752** | 0.687 | 0.648 | 0.559 | 0.788 | 0.706 | 0.793 |
| UN-SAMMEDIA'25 | ✓ | fully | 0.482 | 0.477 | 0.656 | 0.728 | 0.792 | 0.581 | 0.574 | 0.734 | **0.782** | 0.795 |
| Med-SAMEDIA'25 | ✓ | fully | 0.511 | 0.493 | 0.679 | 0.727 | 0.772 | 0.565 | 0.564 | 0.731 | 0.772 | 0.806 |
| COINICCV'25 | ✓ | unsupervised† | 0.519 | 0.410 | 0.581 | 0.706 | 0.748 | 0.572 | 0.515 | 0.690 | 0.746 | 0.778 |
| **SPROUT** | ✓ | training-free† | **0.621** | **0.601** | **0.817** | 0.736 | **0.795** | **0.662** | **0.616** | **0.796** | 0.774 | **0.821** |

⋆ indicates baseline evaluation of the foundation models, and the supervision column specifies the input prompt type. † denotes no ground truth used.

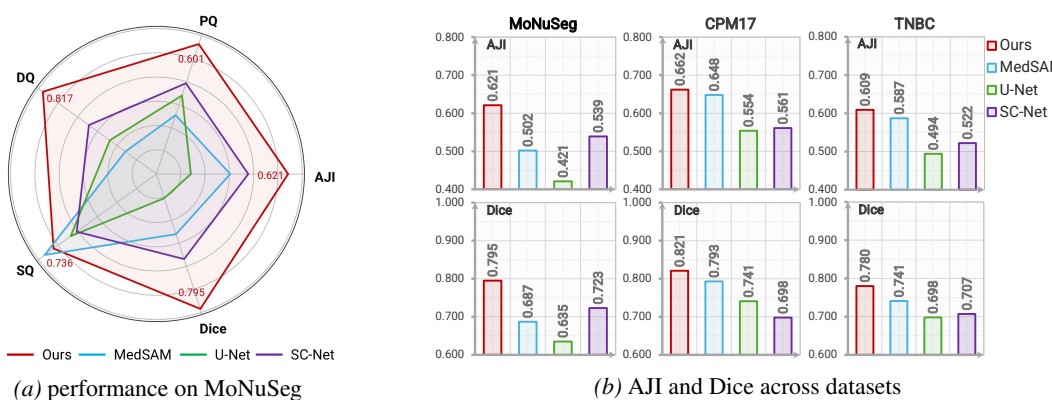

*(a)* performance on MoNuSeg  *(b)* AJI and Dice across datasets

*Figure 3.* **Performance comparison across supervision types.** SPROUT consistently outperforms SAM-based (MedSAM), fully supervised (U-Net), and point-supervised (SC-Net) models across datasets with superior effectiveness in segmentation.

points are uniformly sampled from expanded background masks with an additional stride to ensure sufficient nuclear coverage. The process terminates once multiple compact regions merge into a large connected component, as further expansion risks conflating distinct nuclei. This balances robust assignments with the gradual inclusion of difficult features, resulting in stable and informative prompts.

### 3.3. Instance Mask Prediction and Refinement

The final step is to generate instance-level nuclear masks from activation-derived prompts and refine them to correct boundary errors from overlapping cells and weak edges. To capture fine-grained nuclear structures, we perform patch-level inference, allowing SAM to better localize individual nuclei. Positive and negative prompts further help separate closely packed nuclei from surrounding tissue (Detailed illustrations are in Appendix F.1). When a nucleus has multiple positive cues, each positive together with $y$ nearest negatives is provided to SAM within its patch, and highly overlapped predictions are merged. Although weak inter-nuclear boundaries may cause adjacent nuclei to be predicted as a single instance, fragmentation is rare due to their homogeneous interiors. To this end, we introduce *containment-*

*aware non-maximum suppression (NMS)* that penalizes large masks enclosing multiple smaller nuclei. Specifically, we apply a $\mathtt{tanh}$-based decay penalty proportional to the number of contained instances and combine SAM's confidence with the normalized hematoxylin-channel response into a unified score $S = S_{\text{SAM}} + S'_H$ for filtering. This complementary strategy leverages both morphological consistency and biological priors, suppressing false positives and improving boundary delineation. Implementation details of containment-aware NMS are provided in Appendix E.7.

## 4. Experiments and Results

We evaluate SPROUT on four benchmark datasets, MoNuSeg (Kumar et al., 2017), CPM17 (Vu et al., 2019), TNBC (Naylor et al., 2018), and PanNuke (Gamper et al., 2020) using instance-level metrics (AJI, PQ, DQ, SQ) and the semantic-level Dice coefficient. Complete dataset statistics and metric definitions are provided in Appendix B.1 and B.2. Section 4.1 reports comparisons with representative nuclear instance segmentation methods. Section 4.2 (with Appendix E) presents ablations that validate the key components of SPROUT. We further analyze hyperparameter sensitivity studies in Section 4.3. Additional implemen-

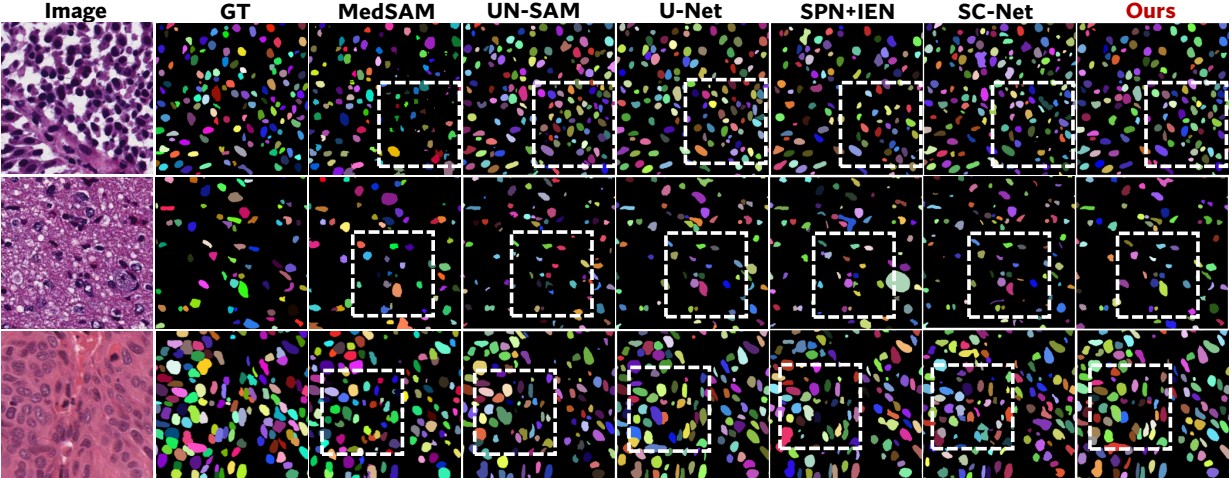

*Figure 4.* **Visualization of instance segmentation results from different methods.** SPROUT delivers more correct instances with fewer overlaps. The highlighted regions show distinct differences.

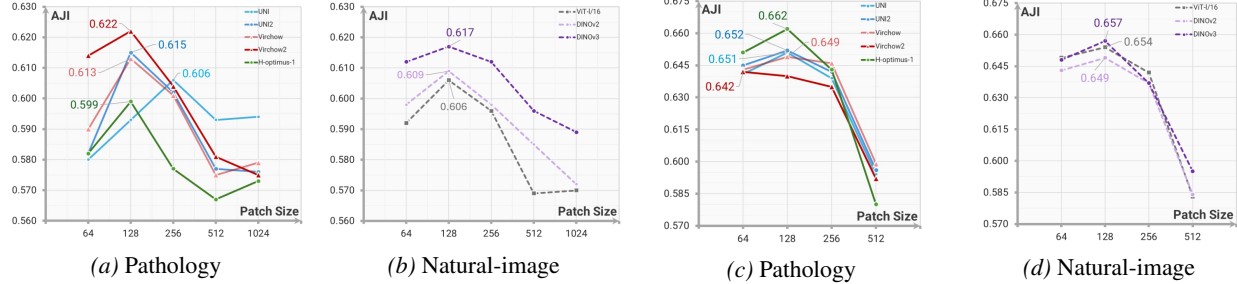

| *(a)* Pathology | *(b)* Natural-image | *(c)* Pathology | *(d)* Natural-image |

*Figure 5.* **Performance comparison of pathology-based and natural-image-based backbones.** On MoNuSeg (a, b) and CPM17 (c, d), the self-reference mask strategy mitigates the domain gap and achieves the best AJI at patch size $128 \times 128$ matching nuclear scale.

tation details are included in Appendix B.3.

## 4.1. Main Results

**Comparison with Baselines.** We benchmark our method against representative and widely used approaches in nuclear instance segmentation, including task-specific architectures (U-Net (Ronneberger et al., 2015), SPN+IEN (Liu et al., 2022), SC-Net (Lin et al., 2023)) and foundation-model-based baselines with domain adaptation or finetuning (SAM (Kirillov et al., 2023), MedSAM (Ma et al., 2024), UN-SAM (Chen et al., 2025b), DES-SAM (Huang et al., 2024a), Med-SA (Wu et al., 2025), and COIN (Jo et al., 2025)). As summarized in Table 1, our method achieves the highest AJI and Dice scores and consistently outperforms all counterparts with up to $8.2\%$ absolute gains in AJI on the challenging MoNuSeg dataset. Notably, the competitive PQ on both datasets demonstrates its strong ability to maintain object-level consistency. We further report comparisons with pathology-specific pretrained models in Appendix D.1 as upper-bound references. Baseline descriptions and reproduction details are in Appendix B.6.

**Analysis by Supervision Type and Visualization.** Figure 3 shows our method surpasses MedSAM, U-Net, and

SC-Net from different supervision regimes without annotations and training. We further make a visual comparison in Figure 4. SPROUT produces clean, non-overlapping masks in challenging cases with nuclei-tissue color similarity or light stain. By automatically generating positive–negative prompts, SPROUT fully exploits SAM's capacity and demonstrates strong robustness across datasets and background appearances. Meanwhile, it offers an efficient alternative to supervised approaches, enabling training-free and annotation-free methods on pathological images. Additional qualitative examples across datasets and visualizations of natural-image segmentation models are provided in Appendix C.1 and C.2.

## 4.2. Ablation Studies

We conduct ablation studies to evaluate the core components of SPROUT, focusing on two key questions: (i) How does SPROUT generate reliable point prompts for SAM prediction effectively? (ii) Which post-processing strategy ensures accurate instance mask selection?

**Feature Extractors.** To assess the effect of the proposed self-reference mask strategy, we evaluate feature extractors trained on both pathology (UNI, UNI2 (Chen et al.,

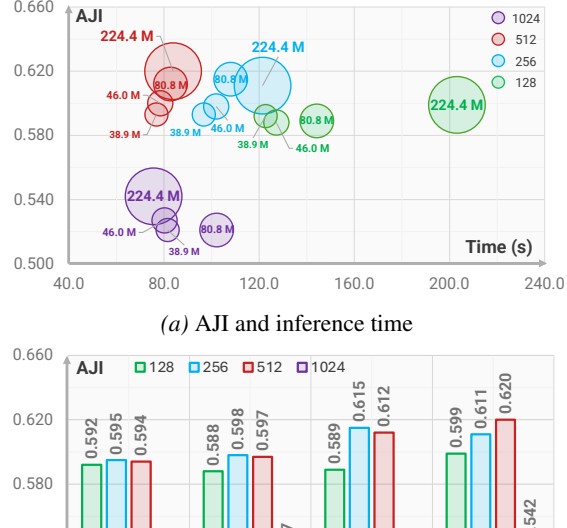

*(a)* AJI and inference time

*(b)* Different image patch sizes

*Figure 6.* **Relationship between SAM variants and patch size.** Appropriate patch sizes narrow the performance gap between large and small SAM variants and enable flexible deployment across resource settings. The best performance is obtained with the large SAM at patch size $512 \times 512$. Results are reported on the MoNuSeg dataset with a fixed patch overlap ratio of $0.5$.

2024), Virchow (Vorontsov et al., 2024), Virchow2 (Zimmermann et al., 2024), H-optimus-1 (Bioptimus, 2025)) and natural images (ViT-l/16 (Dosovitskiy et al., 2020), DINOv2 (Oquab et al., 2024), DINOv3 (Siméoni et al., 2025)) using the MoNuSeg and CPM17 datasets. The Appendix B.7 provides further details on each backbone. As shown in Figure 5, both types of backbones yield comparable AJI scores, typically within $1\%$. Virchow2 and DINOv3 achieve the best performance on MoNuSeg, while H-optimus-1 and DINOv3 perform best on CPM17. These findings validate that the proposed self-reference strategy incorporates image-specific H&E color priors to refine scales and allows cross-domain transfer without specialized fine-tuning. The consistent peaks of AJI at a patch size of $128 \times 128$ align with the feature extractor's receptive field relative to cell sizes. Overly large patches ($\geq 512$) dilute nuclear signals and overly small ones risk fragmenting them. Additional cross-dataset results demonstrating the robustness of self-reference are included in Appendix E.2.

**SAM Variants.** Since SPROUT relies on SAM for instance generation, we analyze how model size influences segmentation. Figure 6 reports AJI for large, base-plus, small, and tiny variants under different patch sizes. Splitting images into moderate patches improves AJI relative to whole-image input, while overly small patches provide

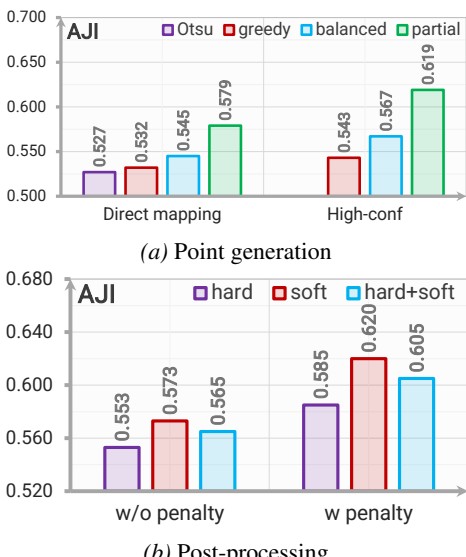

*(a)* Point generation

*(b)* Post-processing

*Figure 7.* (a) High-confidence masks yield cleaner reference features with modest gains. Partial OT outperforms balanced OT by enabling more flexible assignments. (b) Adding a containment-aware penalty consistently improves AJI across NMS types. Soft NMS achieves the best performance by allowing mild overlap, which better matches practical morphology.

little gain and increase computational cost by fragmenting context and amplifying noise. Large and base-plus models perform best, but the advantage over base-plus is minor since nuclei within each patch are relatively homogeneous. Smaller variants also remain competitive with patch inputs, suggesting practical value in resource-limited settings.

**Point Generation.** In Figure 7a, direct Otsu-based separation acts as the color-only baseline and shows its limitation under noise. High-confidence filtering improves robustness by removing unreliable regions. Building on similarity-based feature extraction, greedy mapping yields modest gains, since the locally determined features are easily perturbed by noise. Balanced OT offers marginal improvement by assigning ambiguous pixels to prototypes. In contrast, partial OT delivers the best performance by transporting low-cost, high-confidence matches, leaving uncertain regions in the slack column. See Appendix D.2, E.5, and E.6 for detailed runtime evaluation, comprehensive quantitative results, and point quality analysis.

**Post-processing.** Figure 7b compares NMS strategies with and without the containment-aware penalty. Without the penalty, large masks containing multiple nuclei are retained, as their low IoU with smaller nucleus masks prevents them from being identified during NMS. Introducing the penalty improves AJI by about $5\%$, explicitly discouraging such scenarios. Among selection strategies, soft NMS achieves the best balance by reducing redundant overlaps while still allowing partial overlap in dense regions, consistent with the biological reality of clustered nuclei. The hybrid strategy is

| Image | Prototype 1 | Prototype 2 | Prototype 3 | Activation maps | GT |
|---|---|---|---|---|---|

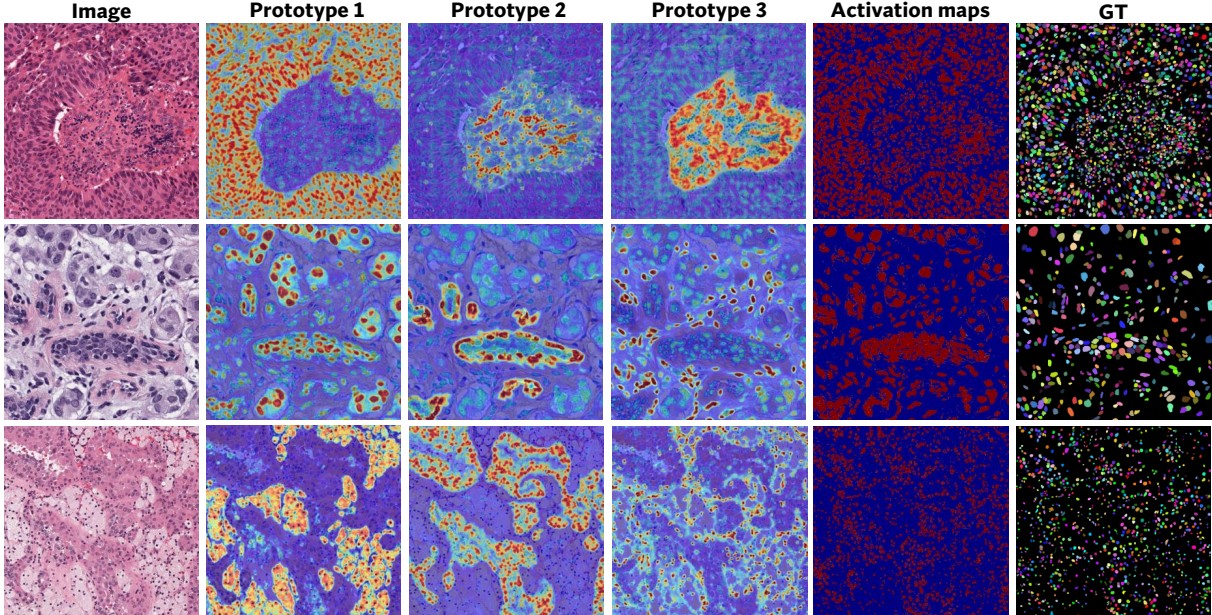

*Figure 8.* **Visualization of class activation (MoNuSeg).** Different prototypes emphasize complementary tissue morphologies, with their aggregation aligning with the ground truth. Clustering-based prototypes capture diverse feature variations for accurate localization.

less suitable because it enforces strict separation rather than acceptable overlaps. Ablations of soft NMS decay functions and score strategies are in Appendix E.7.

**Class Activation.** Figure 8 illustrates the prototype activation after POT-Scan. Each prototype emphasizes distinct morphological patterns, and their combination recovers foreground structures closely aligned with ground truth. This confirms that $K$-means clustering of features produces discriminative prototypes, enabling robust capture of nuclei even under subtle feature diversity.

**Feature Embeddings.** To concretely analyze how POT-Scan reshapes the feature embedding space, we visualize the raw and transported features with UMAP in Figure 9. Before POT-Scan, foreground and background embeddings are highly mixed, showing that raw encoder features alone do not form well-separated clusters under staining variability and weak nuclear boundaries. After POT-Scan, prototype-guided partial transport refines feature assignments, yielding clearer, compact foreground and background groups around prototypes. Features mapped to the slack column lie between the two classes, confirming their ambiguous or noisy nature. As a result, assigning these uncertain features to slack rather than forcing them into either classes improves the reliability of activation maps used for prompt generation.

**Runtime Analysis.** To assess practical deployability, we report runtime characteristics in Table 2. CPM17 and TNBC images are fully processed within 10 seconds, while the more challenging MoNuSeg images require less than three minutes without hardware-level optimization. In real use,

multi-thread parallelism enables concurrent processing. On a single RTX 4090 GPU with four threads, the wall-clock time per image is dramatically reduced. The reported per-nucleus time is measured without parallelization and thus should not be interpreted as practical per-instance latency. Actual throughput is governed by the wall-clock image time, which more accurately reflects the method's deployable performance and can be further shortened by adopting lighter-weight SAM variants. Detailed point-generation runtime is provided in Appendix D.2.

*Table 2.* **Average runtime per image and per nucleus.** Per-image runtime is measured without parallelization. Wall-clock time uses 4-way parallel processing. The H-optimus-1 backbone and large SAM weight are employed.

| Metric | MoNuSeg | CPM17 | TNBC |
|---|---|---|---|
| time/image (s) | 170.651 | 9.452 | 8.305 |
| wall-clock/image (s) | 49.343 | 2.658 | 2.449 |
| time/nucleus (ms) | 282.232 | 78.869 | 103.091 |

### 4.3. Sensitivity Analysis

We analyze the impact of key hyperparameters on segmentation performance to better understand SPROUT's behavior under different configurations. Figure 10 reports results for point generation and mask prediction stages. SPROUT remains stable across all settings, with performance varying by less than 3% except for extreme parameters, indicating minimal hyperparameter tuning in practice. Experimental details are provided in Appendix B.3.

**Point Generation.** In Figure 10a and 10b, the high-

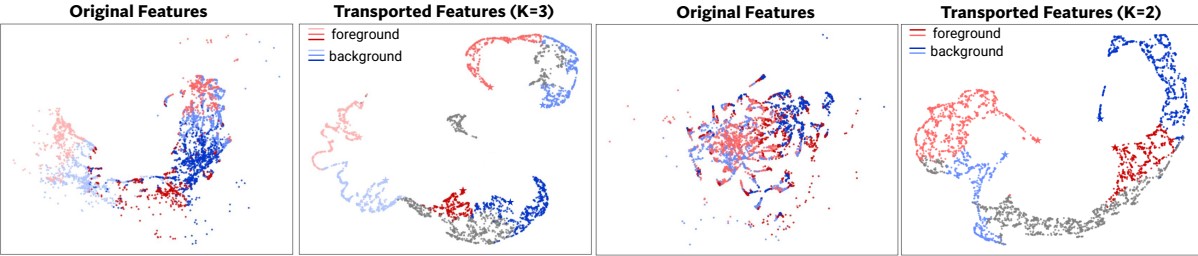

*Figure 9.* **UMAP visualization of feature embeddings before and after POT-Scan.** Red/blue denote foreground/background features, grey denotes slack-assigned features treated as background, and stars mark prototypes. $K$ is the number of cluster centers per class.

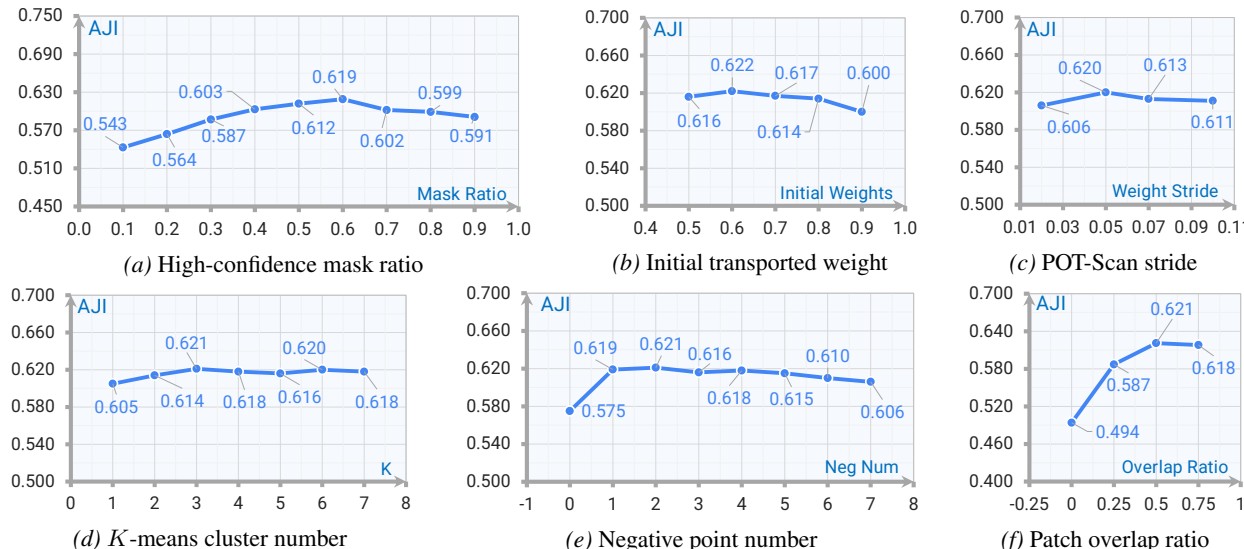

*Figure 10.* **Hyperparameter sensitivity analysis (MoNuSeg).** We evaluate AJI across six representative parameters. Minor variations (less than 3%) are observed under wide changes, confirming SPROUT's robustness to hyperparameter choices with minimal tuning.

confidence mask ratio peaks around $0.6$ and decreases as it approaches 1. This indicates that a moderate ratio balances reliable regions with sufficient coverage, while extreme values either lack generalization or introduce noise. A similar trend is observed for the initial transported weight. The stride used in POT-Scan is relatively milder in Figure 10c. Small strides cause the process to stop early when nuclei are densely packed, while large strides may skip fine details. For the number of $K$-means clusters, small values fail to capture the diversity of foreground and background features. Performance improves as the cluster number increases and stabilizes once $K \geq 3$, as shown in Figure 10d.

**Mask Prediction.** In Figure 10e, introducing a small number of negative points improves accuracy by excluding ambiguous regions that a single positive point cannot separate, as illustrated in Appendix F.1. However, too many negative prompts make the model conservative, leading to smaller and fragmented predictions. The prediction patch overlap ratio has the strongest impact in this stage, as in Figure 10f. Without overlap, AJI is low due to border artifacts. Moderate overlap around $0.5$ is critical for handling boundary regions, while avoiding redundant computation.

## 5. Conclusion

In this work, we presented **SPROUT**, a fully training-free framework for nuclear instance segmentation that generates prompts automatically without annotations. SPROUT introduces a self-reference strategy and a theoretically-grounded POT-Scan scheme to achieve precise feature representation and reduce domain gaps. By guiding SAM with automatically generated point prompts and applying a containment-aware NMS for lightweight refinement, SPROUT yields accurate and efficient segmentation without the need for fine-tuning or adapter modules. Extensive experiments show that SPROUT outperforms previous representative nuclear segmentation methods across multiple challenging datasets and provides new insights into the behavior of prompting-based pipelines. Beyond nuclear segmentation, this work points toward the broader potential of bridging domain gaps via cross-domain priors, offering a path to more robust and adaptable medical imaging models. We acknowledge current limitations, including reliance on SAM for boundary precision and restriction to H&E images, but view SPROUT as a promising, scalable, and trustworthy step toward reliable, end-to-end AI integration in digital pathology.

## Impact Statement

This paper presents work whose goal is to advance the field of Machine Learning. There are many potential societal consequences of our work, none which we feel must be specifically highlighted here.

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

## Table of Contents

## A. Theoretical Analysis

In this section, we provide the theoretical foundations and complete proofs for Section 3.2. We begin with a brief discussion of applications of optimal transport (OT) to set the stage (Section A.1), followed by a review of its standard formulation and the entropic scaling algorithm for efficient computation (Section A.2). We then derive the partial OT formulation adopted in the POT-Scan module by extending from unbalanced OT in Section A.3. In particular, we show that introducing a slack column transforms partial OT into an equivalent unbalanced OT problem, which can be efficiently solved using Sinkhorn-based methods (Section A.4). Finally, we present the algorithm employed to solve partial OT in Section A.5.

## A.1. Background

Optimal Transport (OT) (Villani et al., 2008) provides a mathematical framework for aligning one probability measure with another by finding the most cost-efficient way to reallocate mass. Classical OT corresponds to the case where exact mass preservation is enforced. Variants such as unbalanced OT relax the constraint to handle discrepancies in total mass. Adding entropic regularization (Zhang et al., 2023; Cuturi, 2013) to the objective function enables efficient approximation via Sinkhorn iterations, making large-scale applications practical. Beyond its theoretical elegance, OT has become a versatile tool in modern machine learning. It has supported advances in generative modeling by providing stable training objectives through Wasserstein distances (Gulrajani et al., 2017), in semi-supervised learning by enabling label propagation as a transport problem (Tai et al., 2021), and in domain adaptation by bridging domain shifts through feature distribution alignment (Courty et al., 2016).

## A.2. Scaling Algorithm for Optimal Transport

The optimal transport problem can be formulated as a minimization task over transport plans that determine the cost-optimal way to move mass between two probability measures. Given probability vectors $\mu \in \mathbb{R}^{N \times 1}$, $\nu \in \mathbb{R}^{M \times 1}$, along with a cost matrix $C \in \mathbb{R}_+^{N \times M}$ defined on joint space, the objective function is written as:

$$\min_{T \in \mathbb{R}^{N \times M}} \langle T, C \rangle_F + \phi(T\mathbf{1}_M, \mu) + \psi(T^\top \mathbf{1}_N, \nu) \quad (5)$$

where $T \in \mathbb{R}^{N \times M}$ denotes the transportation plan, $\langle \cdot , \cdot \rangle_F$ is the Frobenius product. $\phi$ and $\psi$ are convex marginal distribution constraints, $\mathbf{1}_M \in \mathbb{R}^{M \times 1}$, $\mathbf{1}_N \in \mathbb{R}^{N \times 1}$ are all one vectors. This is the classical Kantorovich formulation (Kantorovich, 1942) if $\phi$ and $\psi$ are equality constraints. By relaxing the marginal constraints via KL divergence or inequality penalties, the

---

**Algorithm 1** Generalized scaling algorithm

1: **Input:** Cost $C$, regularization $\epsilon > 0$, marginals $\mu \in \mathbb{R}_+^N$, $\nu \in \mathbb{R}_+^M$
2: $Q \leftarrow \exp(-C/\epsilon)$
3: $b \leftarrow \mathbf{1}_n$
4: **while** not converged **do**
5: $\quad x \leftarrow Qb$
6: $\quad \tilde{a} \leftarrow \text{prox}_{\phi/\epsilon}^{KL}(x; \mu)$
7: $\quad a \leftarrow \tilde{a} \oslash x$ ▷ elementwise division
8: $\quad y \leftarrow Q^\top a$
9: $\quad \tilde{b} \leftarrow \text{prox}_{\psi/\epsilon}^{KL}(y; \nu)$
10: $\quad b \leftarrow \tilde{b} \oslash y$
11: **end while**
12: **return** $T^\star = \text{diag}(a)\, Q\, \text{diag}(b)$

---

formulation allows controlled deviations from strict mass preservation. This leads to the unbalanced OT as described in Section A.3.

To make this problem computationally tractable, Cuturi (Cuturi, 2013) proposed entropic regularization. Adding the entropy term $-\epsilon \mathcal{H}(T)$ to objective function leads to the following formulation:

$$\langle T, C \rangle_F - \epsilon \mathcal{H}(T) = \epsilon \langle T, C/\epsilon + \log T \rangle_F = \epsilon \langle T, \log \frac{T}{\exp(-C/\epsilon)} \rangle_F = \epsilon KL(T \| \exp(-C/\epsilon)), \quad (6)$$

Furthermore, Eq.(6) can be reformulated as:

$$\min_{T \in \mathbb{R}_+^{N \times M}} \epsilon KL\left(T \| \exp(-C/\epsilon)\right) + \phi(T\mathbf{1}_M, \mu) + \psi(T^\top \mathbf{1}_N, \nu)\Big). \quad (7)$$

Define the proximal operator as:

$$\text{prox}_{f/\epsilon}^{KL}(y; z) = \arg\min_{x \geq 0} f(x, z) + \epsilon KL(x \| y), \quad (8)$$

where $z$ is the fixed parameter of the function $f$. In our case, $z$ corresponds to the marginal distributions while $f$ represents the associated marginal constraints $\phi$ or $\psi$. Then Eq.(7) can be solved approximately using Alg.(1).

These updates can be interpreted as Bregman projections with respect to the KL divergence onto convex sets defined by the marginal constraints (Benamou et al., 2015). Alternating such projections is guaranteed to converge, and the diagonal scaling form makes each iteration linear in the number of nonzero entries of $Q$. The entropic regularization enforces strict positivity, prevents sparsity and collapse of the transport plan, and enhances numerical stability. Intuitively, the scaling vectors $a, b$ can be viewed as per-row and per-column adjustment factors, respectively. Multiplying by $a$ rescales entire rows

to match $\mu$, while multiplying by $b$ rescales columns to align with $\nu$. The alternating iterations drive the transport plan $T$ to satisfy the marginal structure.

As a result, whenever an optimal transport problem can be reformulated with suitable marginal constraints into the form of Eq.(5), the corresponding proximal operators can be derived as in Eq.(8). This allows the problem to be efficiently solved using Alg.(1).

### A.3. Derivation from Standard OT to Partial OT

When strict equality constraints are not enforced, one may allow mass to be created or discarded. This leads to the unbalanced OT formulation where deviations from the marginals are penalized by a $KL$ divergence. Assuming a uniform source distribution, Eq. (5) can be expressed as:

$$\min_{T \in \Pi} \ \langle T, C \rangle_F + \lambda KL(T^\top \mathbf{1}_N || \frac{1}{M} \mathbf{1}_M)$$
$$\text{s.t. } \Pi = \{T \in \mathbb{R}_+^{N \times M} | \ T\mathbf{1}_M = \frac{1}{N}\mathbf{1}_N\}, \tag{9}$$

where $\lambda$ is the regularization weight factor. Here, the row sums are fixed to the uniform source distribution, while the column sums are softly penalized toward uniformity.

Although unbalanced OT relaxes the marginal constraints, it still penalizes discrepancies between the transported and target mass. As a result, even ambiguous or noisy features are still encouraged to be moved, potentially degrading the quality of the solution. To address this limitation, we adopt the partial OT formulation, which explicitly controls the amount of total transported mass. Instead of hard-thresholding unreliable features, partial OT allows the model to reweigh and selectively transport a subset of the source samples by solving:

$$\min_{T \in \Pi} \ \langle T, C \rangle_F + \lambda KL(T^\top \mathbf{1}_N || \frac{\rho}{M} \mathbf{1}_M)$$
$$\text{s.t. } \Pi = \{T \in \mathbb{R}_+^{N \times M} | T\mathbf{1}_M \leq \frac{1}{N}\mathbf{1}_N, \ \mathbf{1}_N^\top T\mathbf{1}_M = \rho\}, \tag{10}$$

where $N = h \times w$ is the number of source features and $M = 2K$ is the number of target prototypes as described in Eq.(4). $\rho$ specifies the total transported mass and will increase gradually. Intuitively, partial OT still respects the distributional structure but enables progressive selection of reliable samples. Low-cost correspondences are favored first, while noisier or ambiguous features can be safely ignored or deferred until $\rho$ increases. This mechanism provides a principled way to suppress noise while guiding the optimization toward a globally consistent transport plan.

Mathematically, we follow prior work (Caffarelli & McCann, 2010; Chapel et al., 2020; Zhang et al., 2024b) to reformulate the partial OT problem as an unbalanced OT problem that can be solved efficiently with scaling algorithms. The key idea is to introduce a slack column into the marginal distribution to absorb the unselected mass $1 - \rho$, thereby turning the global mass constraint into a marginal one. Specifically, the slack column is denoted as $\eta \in \mathbb{R}^{N \times 1}$ to absorb the remaining mass and form the extended coupling:

$$\hat{T} = [T, \eta] \in \mathbb{R}^{N \times (M+1)}, \quad \hat{C} = [C, \mathbf{0}_N].$$

Imposing row-sum equality to the uniform source and total-mass accounting, we get:

$$\hat{T}\mathbf{1}_{M+1} = \frac{1}{N}\mathbf{1}_N, \quad \mathbf{1}_N^\top \eta = 1 - \rho, \quad \mathbf{1}_N^\top T\mathbf{1}_M = \rho.$$

Thus,

$$\hat{T}^\top \mathbf{1}_N = \begin{bmatrix} T^\top \mathbf{1}_N \\ \eta^\top \mathbf{1}_N \end{bmatrix} = \begin{bmatrix} T^\top \mathbf{1}_N \\ 1 - \rho \end{bmatrix}. \tag{11}$$

Let the target column-mass prior be:

$$\beta = \begin{bmatrix} \frac{\rho}{M}\mathbf{1}_M \\ 1 - \rho \end{bmatrix},$$

we can get the KL-penalized unbalanced surrogate of partial OT as follows:

$$\min_{\hat{T} \in \Phi} \langle \hat{T}, \hat{C} \rangle_F + \lambda KL(\hat{T}^\top \mathbf{1}_N || \boldsymbol{\beta})$$
$$\text{s.t. } \Phi = \{\hat{T} \in \mathbb{R}_+^{N \times (M+1)} | \hat{T}\mathbf{1}_{M+1} = \frac{1}{N}\mathbf{1}_N\}. \tag{12}$$

However, the KL term is soft. Eq.(12) does not guarantee the mass of the last column to be strictly $1 - \rho$. To recover the exact partial-OT constraint, a weighted $KL$ constraint is employed to control the constraint strength for each class:

$$\hat{KL}(\hat{T}^\top \mathbf{1}_N || \beta; \hat{\lambda}) = \sum_{i=1}^{M+1} \lambda_i [\hat{T}^\top \mathbf{1}_N]_i \log \frac{[\hat{T}^\top \mathbf{1}_N]_i}{\beta_i},$$

with

$$\hat{\lambda} = \left[ \begin{array}{c} \lambda \mathbf{1}_M \\ +\infty \end{array} \right].$$

This yields the final equivalent formulation:

$$\min_{T \in \Phi} \langle \hat{T}, \hat{C} \rangle_F + \hat{KL}(\hat{T}^\top \mathbf{1}_N || \beta; \hat{\lambda})$$
$$\text{s.t. } \Phi = \{\hat{T} \in \mathbb{R}_+^{N \times (M+1)} | \hat{T} \mathbf{1}_{M+1} = \frac{1}{N} \mathbf{1}_N\}. \tag{13}$$

The weighted KL makes the slack mass non-negotiable while keeping the real columns softly regularized. So low-cost correspondences are selected first, and ambiguous features can be safely left in the slack. The extended optimal plan is consistent with the original one, and the first $M$ columns of the extended solution align with the optimal plan of the partial OT problem. The proof is provided in the next section.

## A.4. Proof of Equivalence with Partial OT

In this section, we present the full proof that $\tilde{T}^\star$, which is the first M columns of the extended optimal transport plan $\hat{T}^\star$, corresponds exactly to the optimal plan $T^\star$ of the partial OT problem.

*Proof.* Assume the optimal extended plan is:

$$\hat{T}^\star = [\tilde{T}^\star, \eta^\star] \in \mathbb{R}^{N \times (M+1)}, \tilde{T}^\star \in \mathbb{R}^{N \times M}.$$

The weighted KL penalty expands as:

$$\hat{KL}(\hat{T}^{\star\top} \mathbf{1}_N || \beta; \hat{\lambda}) = \sum_{i=1}^{M} \lambda_i [\tilde{T}^{\star\top} \mathbf{1}_N]_i \log \frac{[\tilde{T}^{\star\top} \mathbf{1}_N]_i}{\beta_i} + \lambda_{M+1} \eta^{\star\top} \mathbf{1}_N \log \frac{\eta^{\star\top} \mathbf{1}_N}{1 - \rho}$$
$$= \lambda KL(\tilde{T}^{\star\top} \mathbf{1}_N || \frac{\rho}{M} \mathbf{1}_M) + \lambda_{M+1} \eta^{\star\top} \mathbf{1}_N \log \frac{\eta^{\star\top} \mathbf{1}_N}{1 - \rho}.$$

Taking the limit $\lambda_{M+1} \to +\infty$ forces the slack column to satisfy $\eta^{\star\top} \mathbf{1}_N = 1 - \rho$, otherwise the objective would diverge.

By construction, the extended plan satisfies the row constraint

$$\hat{T}^\star \mathbf{1}_{M+1} = \frac{1}{N} \mathbf{1}_N.$$

This can be written as

$$\tilde{T}^\star \mathbf{1}_M + \eta^\star = \frac{1}{N} \mathbf{1}_N, \quad \eta^\star > 0,$$

we obtain

$$\tilde{T}^\star \mathbf{1}_M \leq \frac{1}{N} \mathbf{1}_N.$$

In addition, the total transported mass of the first $M$ column is

$$\mathbf{1}_N^\top \tilde{T}^\star \mathbf{1}_M = \mathbf{1}_N^\top \hat{T}^\star \mathbf{1}_M - \mathbf{1}_N^\top \eta^\star = 1 - (1 - \rho) = \rho.$$

Therefore,

$$\tilde{T}^\star \in \{\tilde{T}^\star \in \mathbb{R}^{N \times M} | \tilde{T}^\star \mathbf{1}_M \leq \frac{1}{N} \mathbf{1}_N, \mathbf{1}_N^\top \tilde{T}^\star \mathbf{1}_M = \rho\},$$

which is precisely the feasible set of the partial OT problem.

Lastly, the cost of the extended problem is

$$\langle \hat{T}^\star, \hat{C} \rangle_F + \hat{K}L(\hat{T}^{\star\top}\mathbf{1}_N || \boldsymbol{\beta}; \hat{\lambda}) = \langle [\tilde{T}^\star, \eta^\star], [C, \mathbf{0}_n] \rangle_F + \lambda KL(\tilde{T}^{\star\top}\mathbf{1}_N, \frac{\rho}{M}\mathbf{1}_M)$$

$$= \langle \tilde{T}^\star, C \rangle_F + \lambda KL(\tilde{T}^{\star\top}\mathbf{1}_N, \frac{\rho}{M}\mathbf{1}_M)$$

(14)

This is exactly the objective of the partial OT problem as in Eq.(10) evaluated at $\tilde{T}^\star$.

If $\tilde{T}^\star$ achieves a lower cost than $T^\star$ for the initial partial OT formula, it contradicts the optimality of $T^\star$.

If $T^\star$ had strictly lower cost for Eq.(14), then $\tilde{T}^\star$ would no longer achieve the optimum, which would contradict the optimality of $\hat{T}^\star$.

As a result, by convexity of the objective, $\tilde{T}^\star = T^\star$. Dropping the last column of $\hat{T}^\star$, we achieve the optimal transport plan for the partial OT problem. $\qquad\square$

### A.5. Solver for Partial OT

Adding an entropy regularization term $-\epsilon\mathcal{H}(\hat{T})$ to Eq.(13) also enables the efficient scaling algorithm. We denote:

$$Q = \exp(-C/\epsilon), \ f = \frac{\lambda}{\lambda + \epsilon}, \ \alpha = \frac{1}{N}\mathbf{1}_N$$

The optimal plan admits the standard scaling form:

$$\hat{T}^\star = \text{diag}(a)\, Q\, \text{diag}(b).$$

*Proof.* As shown in Section A.2, the main step is to compute the proximal operators corresponding to the constraints $\phi$ and $\psi$. To this end, let us first restate the Eq.(13) in a more general form:

$$\min_{T \in \Phi} \epsilon KL(T \| \exp(-C/\epsilon)) + \hat{K}L(T^\top\mathbf{1}_N || \beta; \lambda),$$

$$\text{s.t.} \quad \Phi = \{T \in \mathbb{R}_+^{N \times M} \mid T\mathbf{1}_M = \alpha\}$$

(15)

where $C$ is the cost matrix, $\alpha$ is the source marginal.

The equality constraint $T\mathbf{1}_M = \alpha$ can be expressed as the indicator:

$$\phi(x; \alpha) = \begin{cases} 0, & x = \alpha, \\ +\infty, & \text{otherwise.} \end{cases}$$

---

**Algorithm 2** Scaling algorithm for partial OT

1: Initialize: Cost matrix $C$, $\epsilon$, $\lambda$, $\rho$, $N$, $K$, a large value $\iota$
2: $C \leftarrow [C, \mathbf{0}_N]$,
3: $\lambda \leftarrow [\lambda, ..., \lambda, \iota]^\top$
4: $\beta \leftarrow [\frac{\rho}{M}\mathbf{1}_M^\top, 1 - \rho]^\top$
5: $\alpha \leftarrow \frac{1}{N}\mathbf{1}_N$
6: $b \leftarrow \mathbf{1}_{M+1}$
7: $Q \leftarrow \exp(-C/\epsilon)$
8: $f \leftarrow \frac{\lambda}{\lambda+\epsilon}$
9: **while** $b$ does not converge **do**
10: $\quad a \leftarrow \frac{\alpha}{Qb}$
11: $\quad b \leftarrow (\frac{\beta}{Q^\top a})^{\circ f}$,
12: $\quad T \leftarrow \text{diag}(a)Q\text{diag}(b)$
13: **end while**
14: **return** $T[:, : 2K]$

---

Plugging this into the proximal operator directly gives: $\text{prox}_{\phi/\epsilon}^{KL}(y; \alpha) = \alpha$.

For the weighted KL penalty, the proximal operator is defined as:

$$\text{prox}_{\psi/\epsilon}^{KL}(y; \beta) = \arg\min_{x \geq 0} \ \hat{K}L(x||\beta; \lambda) + \epsilon KL(x||y)$$

$$= \arg\min_{x \geq 0} \sum_{i=1}^{M+1} \lambda_i(x_i \log \frac{x_i}{\beta_i} - x_i + \beta) + \epsilon(x_i \log \frac{x_i}{y_i} - x_i + y_i).$$

After dropping constants independent of $x$ and regrouping terms, we obtain:

$$\text{prox}_{\psi/\epsilon}^{KL}(y; \beta) = \arg\min_{x \geq 0} \sum_{i=1}^{M+1} (\lambda_i + \epsilon)x_i \log x_i - (\lambda_i \log \beta_i + \lambda_i + \epsilon \log y_i + \epsilon)x_i.$$

Consider the generic function $g(x) = ax \log x - bx$ with $a > 0$,

its derivative is $g'(x) = a(1 + \log x) - b$, hence the minimizer is $x^\star = \exp(\frac{b-a}{a})$. Applying this result gives:

$$x_i^\star = \exp\left(\frac{\lambda_i \log \beta_i + \epsilon \log y_i}{\lambda_i + \epsilon}\right) = \beta_i^{\frac{\lambda_i}{\lambda_i + \epsilon}} \, y_i^{\frac{\epsilon}{\lambda_i + \epsilon}}.$$

In vector notation, we write:

$$x^\star = \beta^{\circ f} \, y^{\circ(1-f)}, \quad f = \frac{\lambda}{\lambda + \epsilon},$$

where $\circ$ denotes the element-wise power.

Now, substituting the two proximal operators into the general scaling algorithm yields the updates:

$$a \leftarrow \frac{\alpha}{Qb}, \quad b \leftarrow \left(\frac{\beta}{Q^\top a}\right)^{\circ f},$$

where $Q = \exp(-C/\epsilon)$.

$\square$

The pseudo-code of the scaling algorithm for partial OT is provided in Alg.(2).

# B. Additional Implementation Details

## B.1. Datasets

We conduct experiments on four challenging benchmark datasets of H&E stained histopathology images: MoNuSeg (Kumar et al., 2017), CPM17 (Vu et al., 2019), TNBC (Naylor et al., 2018), and PanNuke (Gamper et al., 2020).

**(I) MoNuSeg.** MoNuSeg is a multi-organ nuclei segmentation dataset created from H&E-stained tissue images at $40\times$ magnification from the TCGA archive (Weinstein et al., 2013), containing 51 images at $1000 \times 1000$ resolution from 7 organs with a total of 30,837 individually annotated nuclear boundaries. The MoNuSeg dataset contains 37 training images and 14 testing images.

**(II) CPM17.** The CPM17 dataset contains 64 H&E stained histopathology images at $500 \times 500$ resolution with 7,670 annotated nuclei. Each image was scanned at $40\times$ magnification. CPM17 includes 32 images each for training and testing.

**(III) TNBC.** TNBC includes a collection of 50 H&E stained histopathology images of size $500 \times 500$ from Triple Negative Breast Cancer (TNBC) patients, containing 4,028 detailed nuclear annotations.

**(IV) PanNuke.** The PanNuke dataset consists of 7,901 H&E-stained image patches, each sized $256 \times 256$, spanning 19 different organs. In total, it provides 189,744 annotated nuclei. The dataset is partitioned into 2,656 training images, 2,523 validation images, and 2,722 testing images.

To facilitate patch-based processing for both feature extraction and SAM prediction, we use Lanczos interpolation (Lanczos, 1964) over $8 \times 8$ neighborhood to resize the MoNuSeg images to $1024 \times 1024$, and the CPM17 and TNBC images to $512 \times 512$. PanNuke images remain unchanged.

## B.2. Metrics

Nuclear instance segmentation performance is evaluated using four instance-level metrics, including Detection Quality (DQ), Segmentation Quality (SQ), Panoptic Quality (PQ), and the Aggregated Jaccard Index (AJI), along with one semantic-level metric, the Dice coefficient (Dice). The detailed definitions are as follows:

Let $\mathcal{G} = \{G_i\}_{i=1}^{N_G}$ and $\mathcal{P} = \{P_j\}_{j=1}^{N_P}$ denote the sets of ground-truth and predicted instances, respectively. Define the IoU (Intersection over Union) as below:

$$\text{IoU}_{ij} = \frac{|G_i \cap P_j|}{|G_i \cup P_j|}.$$

**Dice.** Dice score measures overall pixel-level agreement and is insensitive to instance identities. It can be calculated using foreground overlap after binarization:

$$\text{Dice} = \frac{2 \left| (\cup_i G_i) \bigcap (\cup_j P_j) \right|}{\left| \cup_i G_i \right| + \left| \cup_j P_j \right|}.$$

**AJI (Aggregated Jaccard Index).** AJI is an instance-aware Jaccard that penalizes both split and merged instances since the unmatched regions go to the denominator.

Let $\mathcal{M} \subseteq \{1, \ldots, N_G\} \times \{1, \ldots, N_P\}$ be a one-to-one matching between instances with $\text{IoU}_{ij} > 0$. Then AJI can be calculated as:

$$\text{AJI} = \frac{\sum_{(i,j) \in \mathcal{M}} |G_i \cap P_j|}{\sum_{(i,j) \in \mathcal{M}} |G_i \cup P_j| + \sum_{i:\, (i,\cdot) \notin \mathcal{M}} |G_i| + \sum_{j:\, (\cdot,j) \notin \mathcal{M}} |P_j|}.$$

**PQ, DQ, and SQ.** Detection Quality (DQ) quantifies the accuracy of instance detection by accounting for false positives and false negatives, while Segmentation Quality (SQ) measures the delineation accuracy of correctly matched instances. Panoptic Quality (PQ), defined as the product of DQ and SQ, provides a comprehensive metric that jointly reflects both detection and segmentation performance.

The matched pairs (true positive) are defined with a fixed threshold $\tau = 0.5$.

$$\text{TP} = \big\{ (i, j) : \text{IoU}_{ij} > \tau \big\}.$$

Let $\text{FP} = N_P - |\text{TP}|$ and $\text{FN} = N_G - |\text{TP}|$. Then

$$\text{DQ} = \frac{|\text{TP}|}{|\text{TP}| + \frac{1}{2}\text{FP} + \frac{1}{2}\text{FN}}, \quad \text{SQ} = \frac{1}{|\text{TP}|} \sum_{(i,j) \in \text{TP}} \text{IoU}_{ij}, \quad \text{PQ} = \text{DQ} \times \text{SQ}.$$

In practice, the one-to-one correspondences between ground truth and predicted instances are established using a greedy matching strategy. The evaluation metrics are subsequently obtained by averaging results across all images.

### B.3. Implementation

**General Settings.** Our method is training-free and therefore does not require dataset partitioning. To ensure a fair comparison with the trained networks, we conduct evaluations on the fixed test sets of MoNuSeg and CPM17. For TNBC, we randomly split the data with an 80/20 ratio for training and testing. The PanNuke dataset partition follows the predefined folds provided in its official release. All experiments are performed on a single NVIDIA GeForce RTX 4090 GPU with 24 GB of memory. As the method is training-free and does not involve stochastic training, performance variations across runs are minimal. We therefore repeat each experiment three times and report the mean performance for clarity.

**Default Implementation Configuration** For clarity and reproducibility, we summarize the default implementation configuration of SPROUT in Table 3.

*Table 3.* **Default implementation configuration.**

| Stage | Setting |
|---|---|
| Self-reference | Mask ratio: 0.6 |
| Feature extraction | UNI2; patch size: $128 \times 128$; overlap ratio: 0.5 |
| Prototype construction | $K$-means; $K = 3$ |
| POT-Scan | Initial mass: 0.6; stride: 0.05 |
| Prompting | Positive: watershed; negative: grid-uniform; nearest 3 |
| SAM inference | SAM-Large; patch size: $512 \times 512$; overlap ratio: 0.5 |
| Post-processing | Soft-NMS; exponential decay; H response + SAM confidence |

**Baseline Reproduction.** For baseline model comparison, we use the official source code released by the authors on GitHub and follow the recommended configurations in the original publications. U-Net predictions are converted into instance

segmentation masks using the watershed algorithm. For baselines that require user input, prompts are generated from the ground truth when no automatic prompt generation mechanism is available. Note that we employ automatic grid-point prompting rather than ground-truth point prompts for the vanilla SAM model. All reported numbers correspond to our reproduced results.

## B.4. Segmentation Performance Summary

We summarize the representative segmentation performance achieved by SPROUT on each dataset in Table 4 to provide an overall view of performance consistency across datasets. All experiments are conducted with the feature extraction resolution fixed at $128 \times 128$, with the large SAM model employed for prediction. The initial mask ratio and initial transport ratio are both set to 0.6, with the POT-Scan weight set to 0.05. We use three $K$-means centers for prototype construction, one positive point and three negative points are provided, and the patch overlap is set to 0.5.

*Table 4.* **Performance summary of SPROUT across datasets.**

| Dataset | Backbone | AJI | PQ | DQ | SQ | Dice |
|---------|----------|-----|-----|-----|-----|------|
| MoNuSeg | Virchow2 | 0.621 | 0.601 | 0.817 | 0.736 | 0.795 |
| CPM17 | H-optimus-1 | 0.662 | 0.616 | 0.796 | 0.774 | 0.821 |
| TNBC | H-optimus-1 | 0.609 | 0.626 | 0.796 | 0.787 | 0.780 |
| PanNuke | H-optimus-1 | 0.581 | 0.534 | 0.700 | 0.763 | 0.766 |

**Sensitivity Studies.** Sensitivity studies are conducted on the MoNuSeg dataset. Unless otherwise specified, all experiments follow the default configuration in Table 3, using Virchow2 features and large SAM weights. We vary one factor at a time. Unless the prompt setting is varied, each SAM prediction uses one positive point and two negative points.

## B.5. Illustrative Visualization

To provide a conceptual understanding of stain priors as discussed in Section 3.1, we illustrate in Figure 11 how hematoxylin and eosin channels naturally separate nuclei from surrounding tissue, producing two complementary intensity maps that act as reliable cues for locating foreground and background regions. This inherent contrast structure allows us to obtain location priors directly from the image without any manual annotations. By overlapping the high-confidence masks derived from the strongest optical density responses with the encoded feature map, we focus on the most concrete and representative nuclear and non-nuclear areas for the following prototype construction. This yields clean and reliable reference features that anchor downstream clustering and similarity mapping rather than arbitrary or heuristic choices.

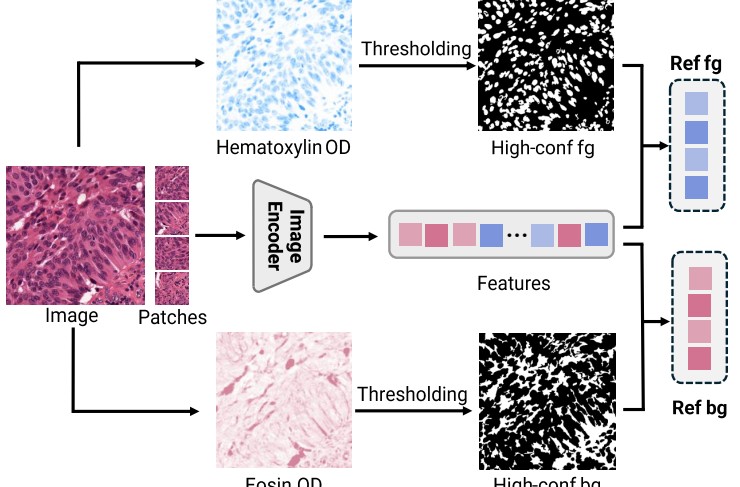

*Figure 11.* **Visualization of stain prior guided features extraction for self-reference.** Hematoxylin and eosin intensity maps from optical density decomposition are thresholded to produce high-confidence foreground and background masks, which are applied to the encoded feature map to extract reference features for subsequent prototype construction.

## B.6. Baselines

We provide detailed descriptions of the baseline models used for comparison in Section 4.1. Their implementation settings and necessary adaptations are provided in Section B.3.

**U-Net.** U-Net (Ronneberger et al., 2015) is a classic convolutional neural network architecture for biomedical image segmentation. Its encoder–decoder design combines a contracting path for contextual extraction with an expansive path for precise localization through symmetric skip connections. This structure allows U-Net to learn fine-grained spatial details from limited training data, making it a widely adopted baseline for medical segmentation tasks.

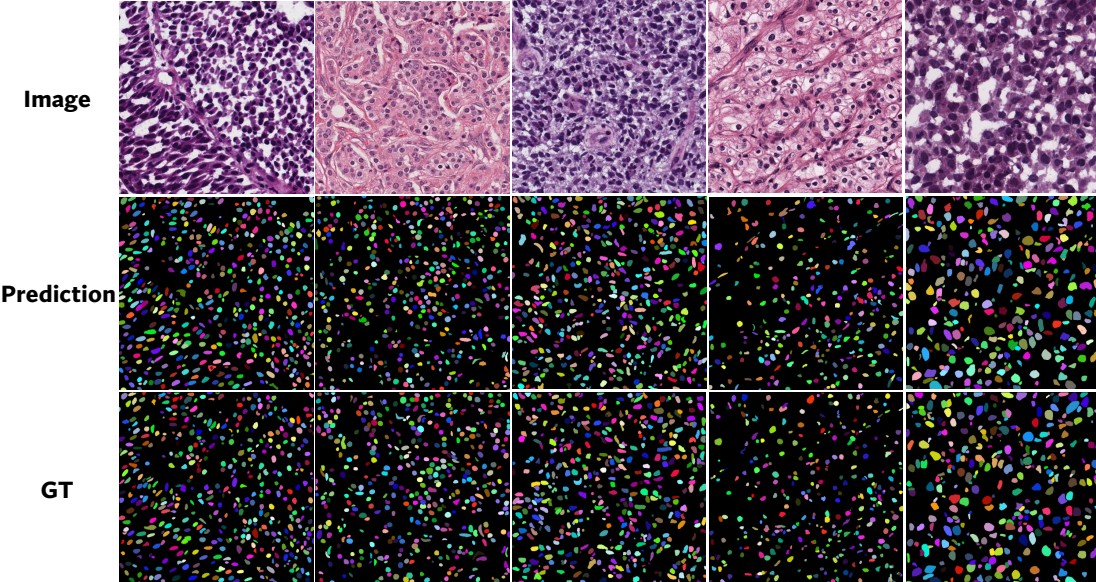

*Figure 12.* **Visualization of segmentation results on MoNuSeg.** Our method accurately distinguishes nuclear regions from surrounding white and stained tissues, and maintains robustness in dense and small nucleus scenarios. Each image may contain thousands of nuclei, and SPROUT consistently identifies and segments them with high fidelity.

**SPN+IEN.** SPN+IEN (Liu et al., 2022) is a weakly supervised framework that uses point annotations for nuclei segmentation. It separates the task into two complementary modules: a Semantic Proposal Network (SPN) that generates coarse foreground–background masks, and an Instance Encoding Network (IEN) that learns instance-aware pixel embeddings to distinguish neighboring nuclei. Such a design reduces annotation costs while maintaining strong segmentation performance.

**SC-Net.** Shape-Constrained Network (SC-Net) (Lin et al., 2023) integrates morphological shape priors into the learning process for nuclei instance segmentation. It employs a detection branch to localize nuclei and a segmentation branch to refine masks. Shape constraints are enforced to better handle overlapping or irregularly shaped nuclei.

**Segment Anything Model.** SAM (Kirillov et al., 2023) is a foundation segmentation model designed for interactive instance segmentation. It consists of an image encoder, a prompt encoder, and a lightweight mask decoder. By leveraging spatial or textual prompts (points, boxes, masks), SAM produces high-quality predictions in a zero-shot manner across diverse domains.

**DES-SAM.** DES-SAM (Huang et al., 2024a) is a distillation-enhanced adaptation of SAM for box-supervised nucleus segmentation. It incorporates a lightweight detection module to generate bounding-box prompts and uses a self-distillation prompting strategy to leverage SAM's pretrained knowledge while fine-tuning a small number of parameters. DES-SAM introduces an edge-aware loss to refine boundary quality as well. Together, these components allow DES-SAM to achieve accurate segmentation with limited supervision while preserving SAM's strong generalization ability.

**MedSAM.** MedSAM (Ma et al., 2024) adapts the SAM to the medical domain and serves as a foundation model for universal medical image segmentation. It is trained on an unprecedented dataset of over 1.5 million image–mask pairs, spanning 10 imaging modalities and over 30 cancer types. In this way, it captures a wide spectrum of anatomical structures and pathological conditions. Like SAM, MedSAM is also a promptable segmentation system, where the user inputs bounding boxes to guide the delineation of regions of interest.

**UN-SAM.** UN-SAM (Chen et al., 2025b) introduces a domain-adaptive self-prompting framework for nuclei segmentation. To remove the manual annotations, it employs a self-prompt generation module to produce high-quality segmentation hints automatically. It further strengthens cross-domain generalization by combining shared representations with domain-specific adaptations, allowing robust performance on heterogeneous nuclei images.

**Med-SA.** Medical SAM Adapter (Med-SA) (Wu et al., 2025) extends SAM to medical imaging via parameter-efficient fine-tuning, updating only about 2% of its parameters. It introduces two key components: SD-Trans, which adapts SAM to 3D medical data, and the Hyper-Prompting Adapter (HyP-Adpt), which conditions the model on user-provided prompts for

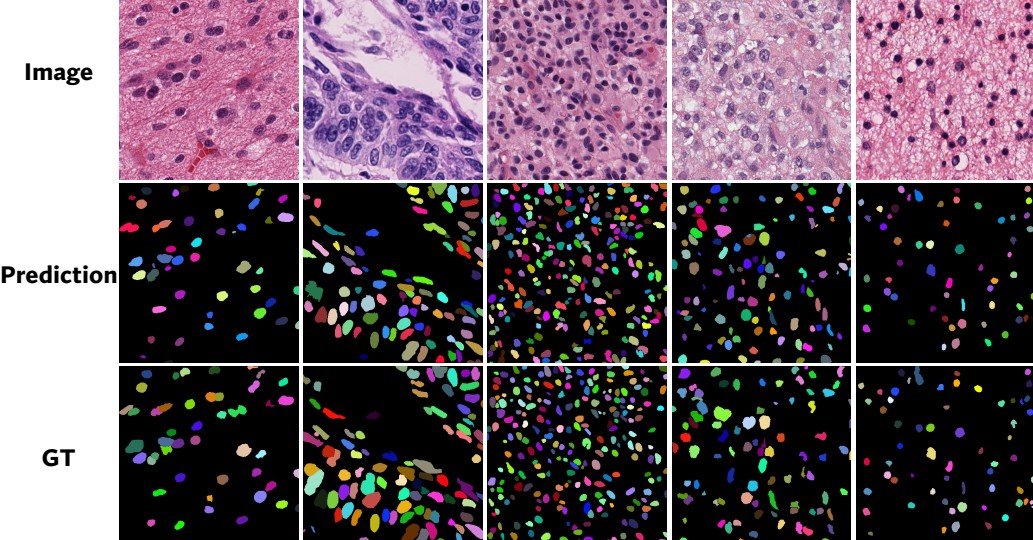

*Figure 13.* **Visualization of segmentation results on CPM17.** SPROUT effectively segments target nuclei in small-scale images when the input exhibits clear foreground–background separation, lightly separated nuclear regions, or tissue and nuclei with similar purple hues that make background discrimination challenging.

interactive segmentation. Med-SA supports both point clicks and bounding box prompts for prediction.

**COIN.** COIN (COnfidence score-guided INstance distillation) (Jo et al., 2025) is an annotation-free framework designed for unsupervised cell instance segmentation. Its three-step pipeline consists of feature extraction with pixel-level propagation, optimal transport–based refinement, SAM-driven instance-level consistency scoring, and recursive self-distillation. It identifies, evaluates, and reinforces reliable pseudo-labels to train a base segmentation network. Overall, COIN offers a simple yet effective strategy for producing high-quality supervisory signals without annotations, introducing a fully unsupervised instance-level confidence scoring paradigm for cell segmentation.

Both COIN and our method operate entirely without annotations. However, COIN begins with a weak initial model and relies on iterative training cycles to strengthen it through distilled supervision. In contrast, our method avoids the training stage altogether by leveraging stain-derived priors to directly obtain high-confidence references for point prompt generation, making it substantially more efficient and faster while preserving strong segmentation quality.

### B.7. Feature Extraction Backbones

**UNI, UNI2.** UNI (Chen et al., 2024) is a pathology-specific vision encoder built with self-supervised pretraining on a large scale. More than 100 million image tiles drawn from around 100,000 diagnostic slides were used. It demonstrates strong transfer across a wide range of pathology tasks, particularly excelling in settings involving rare or underrepresented cancers. UNI-2 enlarges the pretraining corpus to over 200 million H&E and immunohistochemistry (IHC) images from more than 350,000 slides. By incorporating greater scale and modality diversity, UNI-2 further improves generalization across diagnostic tasks. It provides the community with an openly available resource for developing and benchmarking computational pathology models.

**Virchow, Virchow2.** The Virchow model family (Zimmermann et al., 2024; Vorontsov et al., 2024) represents an effort to establish foundation encoders tailored to digital pathology. Virchow was trained in a self-supervised manner on a vast collection of histopathology tiles obtained from millions of whole-slide images (WSIs). This large-scale pretraining equips the model with a broad awareness of tissue architecture and cellular morphology, which can then be transferred to downstream tasks such as cancer subtyping, outcome prediction, or biomarker discovery. Virchow2 extends this approach by further scaling the training corpus to more than 3.1 million WSIs, positioning it among the largest pathology encoders. Beyond its scale, Virchow2 can also serve as a frozen feature extractor for efficient pipeline integration or be fine-tuned to maximize performance on task-specific datasets.

**H-optimus-1.** H-optimus-1 (Bioptimus, 2025) is a 1.1B-parameter vision transformer developed by Bioptimus. It was pretrained with self-supervised learning on billions of histology tiles from over one million slides. It produces rich patch

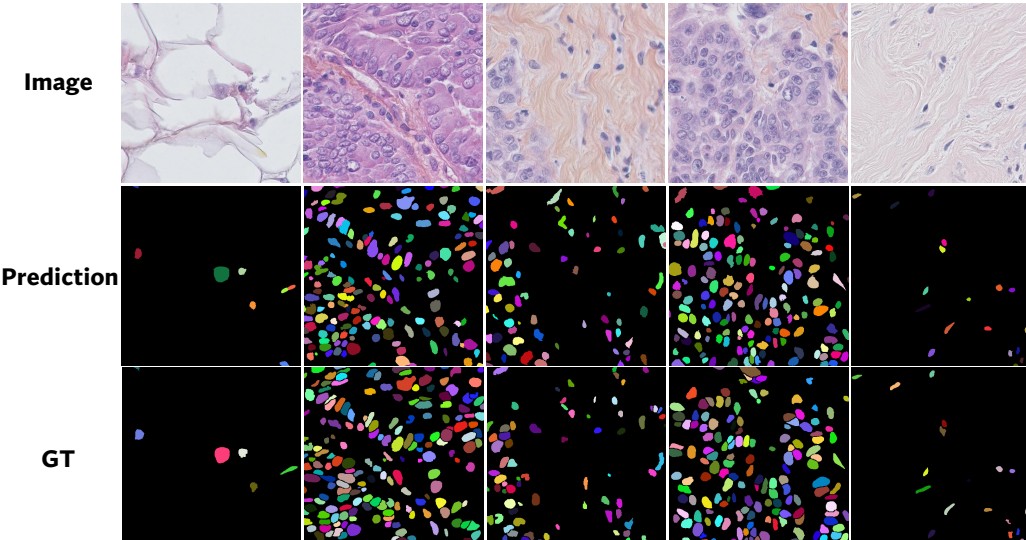

*Figure 14.* **Visualization of segmentation results on TNBC.** Our method remains effective on the TNBC dataset collected under different protocols with an additional transparency channel beyond standard RGB. It performs robustly in both sparse settings with only dozens of nuclei and highly crowded cases with hundreds, accurately distinguishing nuclear regions with minimal omission.

embeddings that capture complex spatial and structural relationships, supporting downstream tasks such as survival analysis, tissue classification, and segmentation.

**DINOv2, DINOv3.** DINOv2 (Oquab et al., 2024) is a self-supervised vision transformer pretrained on a curated collection of 142 million images and is designed to produce general-purpose visual features comparable to those learned by weakly or fully supervised methods. Its representations transfer effectively across domains, enabling broad use without task-specific fine-tuning. DINOv3 (Siméoni et al., 2025) extends DINOv2 by scaling pretraining to over a billion images and models with billions of parameters. It further introduces mechanisms to stabilize training and preserve dense feature quality. With additional distillation into smaller, efficient variants, it achieves superior performance on both recognition and dense prediction tasks, underscoring the role of scale-aware design in vision foundation models.

**ViT-l/16.** The Vision Transformer (ViT) family adapts the Transformer architecture to images by treating fixed-size patches as tokens. Among its configurations, ViT-l/16 (Large, $16 \times 16$ patch size) (Dosovitskiy et al., 2020) has become a widely adopted backbone due to its balance of scale and granularity. It consists of 24 Transformer layers with a hidden dimension of 1024 and roughly 307M parameters. Pretraining on large-scale datasets demonstrates that ViT-l/16 gains substantial improvements from scaling. Its $16 \times 16$ patch size ensures a manageable sequence length while preserving sufficient spatial resolution, making it a widely used backbone and a standard reference model for feature extraction in vision transformers.

### B.8. Additional Algorithms

**Watershed algorithm.** The watershed algorithm (Vincent & Soille, 1991) is a classical segmentation approach for delineating touching or overlapping objects. It operates on the analogy of a topographic surface, where the grayscale image is interpreted as an elevation map. Pixels of lower intensity correspond to basins (valleys), whereas higher intensity values represent ridges (peaks). By flooding this landscape, water initially fills the basins, and as the level rises, neighboring catchment areas begin to merge. To prevent such mergers, separating boundaries are introduced at the points of convergence, thereby yielding distinct object regions. In practice, the algorithm is driven by the inverse of the distance transform, with local maxima serving as initial seeds or markers that guide the flooding process. Within our framework, the centroid of each resulting watershed region is used as the positive point prompt for prompt-based segmentation.

**Otsu's thresholding.** Otsu's thresholding (Otsu et al., 1979) is a global image binarization technique that selects the threshold $t^\star$ by minimizing the intra-class variance of foreground and background pixels, which is equivalent to maximizing the between-class variance.

Suppose the image histogram has $L$ gray levels, with normalized probabilities $p_i$ for each level $i$. For a given threshold $t$, the images are divided into two classes: class 0 with gray level $1, \cdots, t$ with probability $w_0(t) = \sum_{i=1}^{t} p_i$ and mean $\theta_0(t)$,

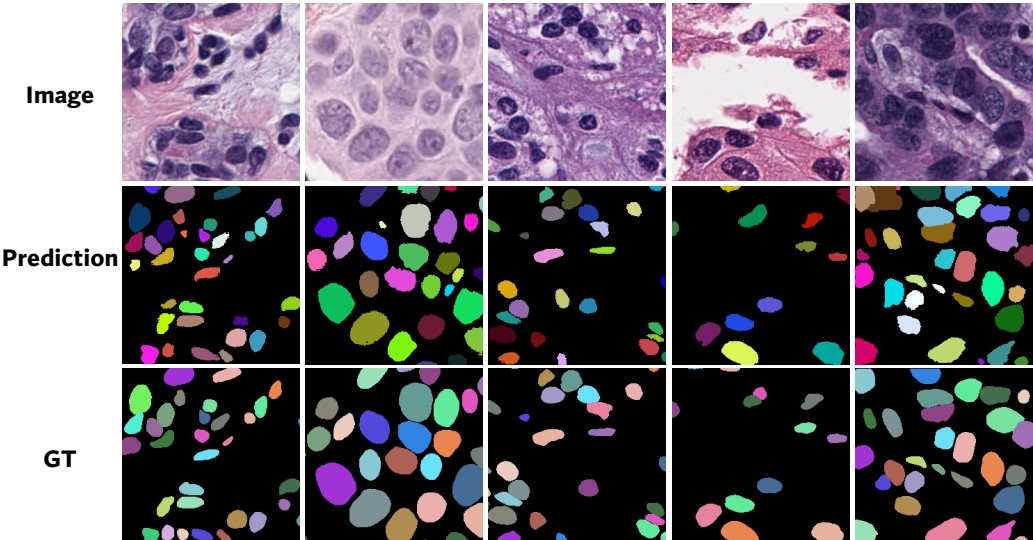

*Figure 15.* **Visualization of segmentation results on PanNuke.** PanNuke contains relatively larger and fewer nuclei with diverse tissue origins. These examples demonstrate the generalizability of our method across diverse nuclear morphologies and stain appearances, despite substantial structural and visual variation.

class 1 with gray level $t + 1 \cdots, L$ with probability $w_1(t) = \sum_{i=t+1}^{L} p_i$ and mean $\theta_1(t)$. The total mean is $\theta_T = \sum_{i=1}^{L} i p_i$. The between-class variance is:

$$\sigma^2(t) = w_0(t)(\theta_0(t) - \theta_T)^2 + w_1(t)(\theta_1(t) - \theta_T)^2$$

Otsu's method chooses the threshold:

$$t^\star = \arg \max_t \sigma^2(t)$$

which maximizes the separation between the two classes. This makes the method non-parametric and unsupervised, requiring no prior knowledge about the image content for foreground–background separation.

## C. Additional Qualitative Results

In this section, we provide the qualitative visualizations of segmentation results from our SPROUT pipeline across four datasets (Section C.1), complementing the analysis in Figure 4. For comparison, we also include results from semantic and instance segmentation models developed for natural images in Section C.2. It helps illustrate the gap between natural and pathological domains and highlights the unique challenges of nuclear image segmentation discussed in Introduction (Section 1).

### C.1. Segmentation Results across Datasets

As illustrated in Figure 12-15, SPROUT delivers accurate nuclear segmentation across diverse datasets. The method demonstrates robustness under varying cellular organizations, from crowded fields containing thousands of nuclei (Figure 12 and Figure 13) to extremely sparse images with only a few (Figure 14). It performs well even when nuclei are small, densely packed, or exhibit subtle color contrasts with surrounding tissues. Moreover, SPROUT adapts seamlessly to the TNBC dataset, which is acquired with different staining protocols and image formats, including those with non-standard channels, consistently producing reliable delineations. It also generalizes robustly to the PanNuke dataset in Figure 15, where the relatively extensive spatial coverage of nuclei often risks over-segmentation. SPROUT preserves coherent nuclear structures without fragmenting any individual cells. Note that both predicted and ground-truth instance masks are displayed with randomly assigned colors, which serve only to differentiate instances for better visualization. Identical colors do not imply correspondence across cells or categories.

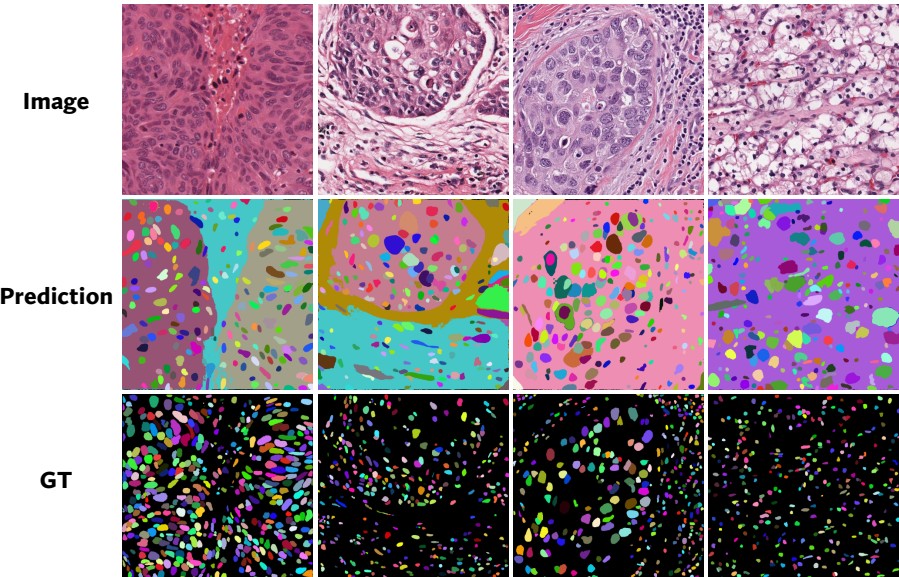

*Figure 16.* **Visualization of instance segmentation results from SAM auto-prompting.** SAM captures a subset of nuclei but struggles to separate nuclei from background tissues, leading to compromised segmentation.

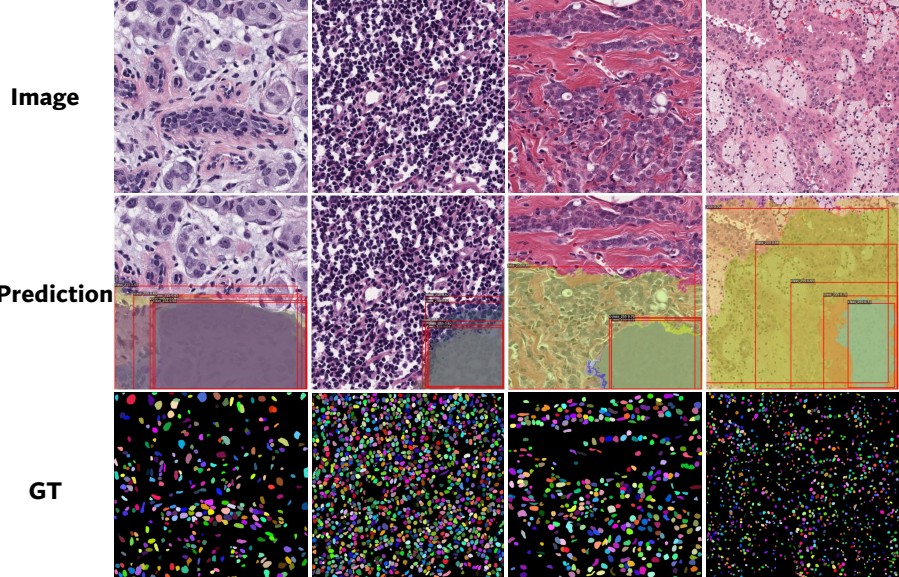

*Figure 17.* **Visualization of segmentation results from Mask DINO.** The model highlights broad tissue regions but fails to resolve individual nuclei.

### C.2. Comparison with Natural Image Segmentation Models

To assess the intrinsic difference between segmentation models developed for natural images and those tailored to pathological images, we apply annotation-free natural image models to nuclear segmentation. Specifically, we choose SAM (Kirillov et al., 2023), MaskDINO (Li et al., 2023), and CutLER (Wang et al., 2023) for instance segmentation and Bridge the Points (Zhang et al., 2024a) for one-shot semantic segmentation. The description of SAM is provided in Appendix B.6, while the remaining models are summarized below.

**SAM.** As shown in Figure 16, SAM auto-prompting operates on a fixed $32 \times 32$ grid, which fails to distinguish foreground from background reliably and cannot ensure coverage of all nuclei. Large tissue regions are often misclassified as targets, which suppresses small-cell predictions either due to lower SAM scores or the lack of point prompts generated on a coarse grid. In addition, because the selection of final masks relies on morphological scores rather than semantic information or

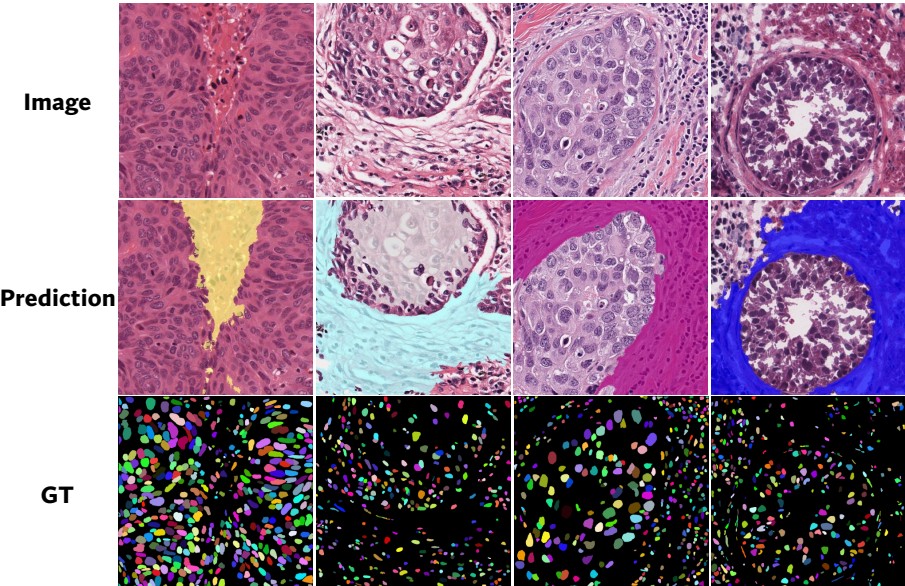

*Figure 18.* **Visualization of segmentation results from CutLER.** The model segments coarse regions with boundary changes but fails to capture fine-grained nuclear structures. The colored areas denote the produced masks.

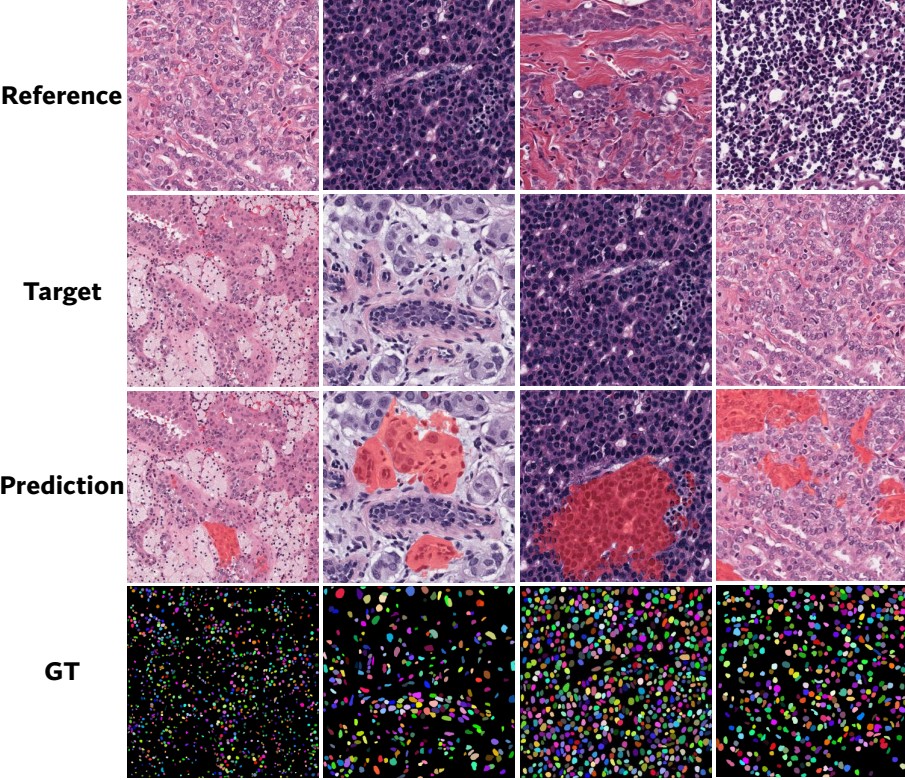

*Figure 19.* **Visualization of semantic segmentation results from Bridge the Points.** Red areas represent the segmented regions. Designed for natural images, the method fails to capture scattered nuclei or distinguish individual cells, causing predictions to focus on a few prominent areas and merge many nuclei together.

negative prompts, the resulting masks frequently exhibit over-segmentation, reflecting SAM's dependence on purely visual cues and the strong similarity between nuclear regions and surrounding background textures.

**Mask DINO.** Mask DINO extends the transformer-based object detector DINO by adding a parallel mask prediction

branch, thereby unifying detection with instance, semantic, and panoptic segmentation through shared query embeddings for bounding box regression and mask generation. Pretrained on large-scale detection and segmentation datasets, it achieves strong performance on natural image benchmarks.

However, as shown in Figure 17, its performance on MoNuSeg is limited. The top predictions frequently overlap on large tissue regions rather than capturing individual nuclei. This behavior suggests a tendency to segment regions of homogeneous texture or color areas, while failing to delineate fine-grained nuclear structures.

**CutLER (Cut-and-Learn).** CutLER is a framework for unsupervised object detection and instance segmentation. It builds on MaskCut, which generates coarse object masks from self-supervised Vision Transformer features, and then refines them through detector training with robust loss dropping and iterative self-training. Operating entirely without human annotations, it provides a strong baseline for natural image segmentation.

In our experiments, only the MaskCut stage is applied to produce instance masks, without training the entire network. Because its affinity matrix relies on feature similarity, which is often weakly distinguishable in pathological images, CutLER primarily segments regions with sharp boundary contrasts but is unable to identify small-scale nuclei, as illustrated in Figure 18. Additionally, many nuclear regions remain undetected due to feature homogeneity.

**Bridge the Points.** Bridge the Points is a graph-based extension of SAM designed for few-shot semantic segmentation. It automatically selects prompts and aligns them with mask granularity through graph connectivity, reducing reliance on hyperparameters and mask refinement. This makes it suitable for low-data settings and cross-domain generalization.

In our experiments, semantic masks from reference images were provided together with random target images in a one-shot configuration. As shown in Figure 19, the method improves the delineation of multiple smaller regions compared to other semantic approaches but still operates largely at the tissue level and fails to resolve nuclei at the instance level. Alongside the results from the instance segmentation baselines, this highlights the limitations of directly applying natural image models to the pathological domain.

## D. Additional Quantitative Study

### D.1. Histopathology-Specific Pretrained Baselines

We additionally benchmark SPROUT against representative large-scale in-domain pretrained models to provide an upper-bound reference in Table 5, including Cellpose (Stringer et al., 2021), CellViT (Hörst et al., 2024), CellSAM (Marks et al., 2025), PathoSAM (Griebel et al., 2025). Overall, SPROUT achieves competitive performance relative to in-domain models with no training or pixel-wise annotations. While pretrained methods benefit from additional in-domain data and specialized mechanisms for boundary delineation, the remaining performance gap remains moderate, highlighting SPROUT as a practical and efficient alternative when training resources or annotations are unavailable.

*Table 5.* **Upper-bound nuclear instance segmentation performance evaluations.** We compare SPROUT with histopathology-specific pretrained fully supervised models on the MoNuSeg and CPM17 datasets to assess the achievable performance upper bound. Segmentation performance is reported using AJI, PQ, DQ, SQ, and Dice. Best results are highlighted in **bold**, and second-best results are underlined.

| Method | SAM | MoNuSeg | | | | | CPM17 | | | | |
|---|---|---|---|---|---|---|---|---|---|---|---|
| | | AJI | PQ | DQ | SQ | Dice | AJI | PQ | DQ | SQ | Dice |
| CellposeNat. Methods'20 | ✗ | 0.627 | 0.611 | 0.824 | 0.742 | **0.806** | 0.691 | 0.653 | 0.841 | **0.777** | 0.827 |
| CellViTMEDIA'24 | ✓ | **0.657** | **0.637** | **0.845** | **0.754** | 0.802 | **0.703** | **0.658** | **0.852** | 0.772 | 0.825 |
| CellSAMNat. Methods'25 | ✓ | 0.606 | 0.610 | 0.815 | 0.749 | 0.788 | 0.660 | 0.638 | 0.822 | 0.776 | 0.829 |
| PathoSAMMIDL'25 | ✓ | 0.316 | 0.116 | 0.186 | 0.621 | 0.533 | 0.671 | 0.623 | 0.808 | 0.771 | **0.832** |
| **SPROUT** | ✓ | 0.621 | 0.601 | 0.817 | 0.736 | 0.795 | 0.662 | 0.616 | 0.796 | 0.774 | 0.821 |

### D.2. Runtime Analysis

To fully assess the computational overhead of our approach in practical settings as a complement to Figure 6 in Section 4.2, we report detailed runtime statistics for the point-generation pipeline. The runtime breakdown for each step in the point-generation pipeline is summarized in Table 6a and 6b. The initial mask stage, which identifies coarse foreground and background regions from stain information, is required for all methods. The prototype stage performs feature extraction and $K$-means clustering to construct reference prototypes. Greedy search assigns features directly to their nearest prototypes

and serves as a baseline against OT.

Otsu's method uses solely the stain-intensity histogram of the input image, operating directly on optical-density values without any feature encoding, clustering, or contextual reasoning. This oversimplified thresholding is fast but leads to coarse separation of nuclear and non-nuclear regions. Greedy assignment provides a stronger baseline by performing a local nearest-prototype lookup for each feature vector. It has a computational complexity of $O(NK)$, where $N$ is the number of feature vectors and $K$ is the number of prototypes. This operation is essentially a single cosine-similarity pass over the encoded feature map. As a result, it is fast but remains inherently local, lacking any global consistency constraint. In contrast, Optimal Transport methods solve a global alignment problem using Sinkhorn iterations, which typically requires $O(T(NK))$ operations, where $T$ is the number of iterations. POT-Scan reformulates partial OT into a series of balanced OT problems and gradually increases the transported mass. Although this process usually converges in 1–3 mass-increase steps per image, each step requires solving an OT problem. Consequently, it is slower than greedy matching or Otsu, but the runtime is still comparable to the feature-extraction stage, which remains the dominant cost across all methods.

*Table 6.* **Point generation runtime breakdown on MoNuSeg and CPM17.** Both tables report runtime in seconds without parallel processing. Feature extraction resolution: $128 \times 128$. Partial OT initialized with weight 0.7 and stride 0.05.

*(a)* **MoNuSeg.** Image size: $1024 \times 1024$.

| Stage | Otsu | greedy | balanced | partial |
|---|---|---|---|---|
| Inital mask | 0.359 | 0.341 | 0.341 | 0.341 |
| Prototype | - | 6.024 | 6.024 | 6.024 |
| Mapping | - | 0.001 | 3.279 | 9.605 |
| Point-gen | 0.137 | 0.137 | 0.137 | 0.137 |
| End-to-end | 0.488 | 6.503 | 9.781 | 16.107 |

*(b)* **CPM17.** Image size: $512 \times 512$.

| Stage | Otsu | greedy | balanced | partial |
|---|---|---|---|---|
| Inital mask | 0.076 | 0.071 | 0.071 | 0.071 |
| Prototype | - | 0.963 | 0.963 | 0.963 |
| Mapping | - | 0.001 | 0.618 | 1.991 |
| Point-gen | 0.027 | 0.027 | 0.027 | 0.027 |
| End-to-end | 0.103 | 1.062 | 1.679 | 3.052 |

Critically, the additional computation of the POT-Scan module is not overhead for its own sake. The partial-mass formulation enables POT to discard ambiguous, low-confidence features while focusing transport on the most reliable structures. This global, noise-aware selection mechanism yields stable and high-quality point prompts, particularly useful in challenging histopathology images where local heuristics often fail. In this context, the computational cost becomes an economically reasonable trade-off for obtaining globally consistent, noise-robust prototype alignment.

Additionally, both feature extraction and OT naturally scale with image resolution. The CPM17 datasets with smaller inputs ($512 \times 512$) exhibit shorter runtime than the high-resolution MoNuSeg dataset, where images are $1024 \times 1024$. Moreover, hyperparameters such as the initial mass and spatial stride offer practical trade-offs between runtime and accuracy. Increasing either parameter reduces computation while incurring only minor performance degradation since the method is insensitive to these hyperparameter settings.

### D.3. Robustness to Image Perturbations

We further evaluate the robustness of SPROUT under different stain normalization choices and moderate stain perturbations. As shown in Table 7, SPROUT achieves stable performance with both Macenko and Vahadane normalization, showing only small differences compared with the original setting. Synthetic perturbations lead to minor performance drops, while optical density scaling is more challenging but does not cause performance

*Table 7.* **Robustness analysis under stain normalization choices and synthetic stain perturbations.** B, C, S, and H denote brightness, contrast, saturation, and hue, respectively.

| Setting | AJI | Dice |
|---|---|---|
| Macenko | 0.619 | 0.799 |
| Vahadane | 0.614 | 0.793 |
| BCS $\pm$ 0.1 | 0.617 | 0.792 |
| BCS $\pm$ 0.2 | 0.611 | 0.790 |
| H $\pm$ 0.02 | 0.616 | 0.791 |
| H $\pm$ 0.05 | 0.612 | 0.789 |
| BCS $\pm$ 0.1, H $\pm$ 0.02 | 0.618 | 0.792 |
| BCS $\pm$ 0.2, H $\pm$ 0.05 | 0.612 | 0.790 |
| H $\times$ 0.8 | 0.604 | 0.782 |
| E $\times$ 0.8 | 0.607 | 0.783 |
| H $\times$ 1.2 | 0.608 | 0.784 |
| E $\times$ 1.2 | 0.609 | 0.785 |
| H $\times$ 0.8, E $\times$ 1.2 | 0.602 | 0.784 |
| H $\times$ 1.2, E $\times$ 0.8 | 0.606 | 0.783 |
| SPROUT | 0.621 | 0.795 |

collapse. These results indicate that SPROUT remains robust under moderate stain variations.

## D.4. Practical Efficiency

To contextualize the practical efficiency of SPROUT, we provide a direct runtime comparison with representative segmentation pipelines on CPM17 in Table 8. Compared with lightweight feed-forward models such as U-Net, SPROUT has higher test-time latency because its inference includes POT-Scan and SAM prediction. However, this cost is incurred without target-dataset annotation, training, or retraining. Compared with MedSAM, SPROUT requires moderately higher inference time but achieves better segmentation performance. COIN is faster at test time, but it has the highest setup cost due to SAM-involved training. Therefore, SPROUT trades increased test-time computation for substantially reduced target-dataset setup cost, while maintaining strong segmentation accuracy without additional annotation or retraining.

*Table 8.* **Practical runtime comparison on CPM17.**

| Method | Train? | Ann.? | New data retrain? | s/img | AJI |
|--------|--------|-------|-------------------|-------|-----|
| U-Net | ✓ | ✓ | ✓ | 0.294 | 0.554 |
| MedSAM | ✓ | ✓ | × | 7.242 | 0.648 |
| COIN | ✓ | × | ✓ | 4.311 | 0.572 |
| SPROUT | × | × | × | 9.452 | 0.662 |

# E. Additional Ablation Study

In this section, we report comprehensive segmentation results on MoNuSeg, CPM17, TNBC, and PanNuke across all feature extraction backbones (Section E.2). The complete sets of evaluation metrics, together with additional ablation studies on segmentation backbones, clustering configuration, point generation, prompt sampling, and post-processing, are provided in Section E.3-E.5 and Section E.7, respectively. In Section E.1, we further evaluate the impact of backbone size to assess the practical applicability of SPROUT under limited-resource settings.

## E.1. Backbone Variants

To assess the applicability of our method under constrained computational budgets, we evaluated feature extractors of different sizes. Since pathology-specific backbones do not provide multiple capacity tiers, we adopted the ViT family for backbone scaling. The results in Figure 20 indicate that AJI degrades only marginally as the backbone becomes smaller. The large and base variants perform comparably, while the small and tiny variants remain within 2% of the larger models. This mild performance drop can be attributed to the relatively low visual complexity of pathology nuclei, where smaller models still capture sufficiently discriminative features. Moreover, the proposed self-reference strategy in our pipeline provides per-image calibration, further stabilizing the quality of features for point generation. This demonstrates the minimal computational trade-off when reducing backbone capacity. Overall, these findings suggest that our method remains highly effective even with lightweight backbones, making it well-suited for deployment in resource-limited settings.

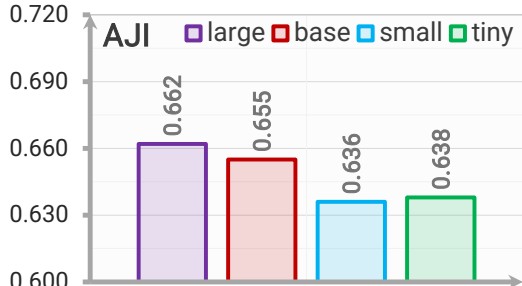

*Figure 20.* **Effects of backbone variants.** Results are reported using Vision Transformer backbones on the CPM17 dataset. Feature extraction resolution: $128 \times 128$. SAM weights: large. SAM prediction patch size: $512 \times 512$.

## E.2. Feature Extractors across Datasets

We present detailed segmentation results on the MoNuSeg, CPM17, TNBC, and PanNuke datasets across different backbones and patch sizes (Table 9–16). The best results are highlighted in **bold**, and the second-best are underlined. These results confirm the effectiveness of our self-reference strategy, showing consistent generalization and comparable performance between natural-image-based and pathology-specific backbones. Unless otherwise specified, all subsequent predictions are obtained using the large SAM model with a patch size of $512 \times 512$ with an overlap ratio of $0.5$. Soft NMS is employed as the post-processing method.

## E.3. Segmentation Backbones

*Table 9.* **Segmentation results of AJI and Dice with different backbones on the MoNuSeg dataset.**

| Backbones | AJI | | | | | Dice | | | | |
|---|---|---|---|---|---|---|---|---|---|---|
| | 64 | 128 | 256 | 512 | 1024 | 64 | 128 | 256 | 512 | 1024 |
| UNI | 0.580 | 0.593 | 0.606 | 0.593 | 0.599 | 0.772 | **0.795** | 0.787 | 0.766 | 0.778 |
| UNI2 | 0.582 | 0.615 | 0.602 | 0.577 | 0.576 | 0.774 | 0.780 | 0.777 | 0.760 | 0.761 |
| Virchow | 0.590 | 0.613 | 0.621 | 0.575 | 0.579 | 0.764 | 0.787 | 0.780 | 0.759 | 0.762 |
| Virchow2 | 0.614 | **0.622** | 0.604 | 0.581 | 0.575 | 0.770 | **0.795** | 0.776 | 0.763 | 0.759 |
| H-optimus-1 | 0.582 | 0.599 | 0.577 | 0.568 | 0.573 | 0.772 | 0.785 | 0.781 | 0.751 | 0.752 |
| ViT-l/16 | 0.592 | 0.606 | 0.596 | 0.569 | 0.570 | 0.764 | 0.784 | 0.768 | 0.760 | 0.758 |
| DINOv2 | 0.598 | 0.609 | 0.598 | 0.582 | 0.572 | 0.770 | 0.784 | 0.777 | 0.764 | 0.756 |
| DINOv3 | 0.612 | 0.617 | 0.612 | 0.596 | 0.589 | 0.779 | 0.789 | 0.780 | 0.766 | 0.773 |

*Table 10.* **Segmentation results of DQ, SQ, and PQ with different backbones on the MoNuSeg dataset.**

| Backbones | DQ | | | | | SQ | | | | | PQ | | | | |
|---|---|---|---|---|---|---|---|---|---|---|---|---|---|---|---|
| | 64 | 128 | 256 | 512 | 1024 | 64 | 128 | 256 | 512 | 1024 | 64 | 128 | 256 | 512 | 1024 |
| UNI | 0.752 | 0.805 | 0.793 | 0.774 | 0.772 | 0.730 | 0.731 | 0.733 | 0.729 | 0.723 | 0.549 | 0.588 | 0.581 | 0.564 | 0.558 |
| UNI2 | 0.751 | 0.803 | 0.792 | 0.762 | 0.769 | 0.732 | 0.733 | 0.733 | 0.730 | 0.726 | 0.550 | 0.589 | 0.581 | 0.556 | 0.558 |
| Virchow | 0.749 | 0.809 | 0.792 | 0.756 | 0.770 | 0.733 | 0.732 | 0.732 | 0.729 | 0.727 | 0.549 | 0.592 | 0.580 | 0.551 | 0.560 |
| Virchow2 | 0.785 | **0.817** | 0.803 | 0.770 | 0.762 | 0.734 | 0.736 | 0.733 | 0.731 | 0.728 | 0.576 | **0.601** | 0.589 | 0.563 | 0.555 |
| H-optimus-1 | 0.767 | 0.796 | 0.795 | 0.762 | 0.763 | 0.733 | 0.733 | 0.735 | 0.729 | 0.726 | 0.562 | 0.583 | 0.584 | 0.555 | 0.554 |
| ViT-l/16 | 0.764 | 0.797 | 0.785 | 0.751 | 0.762 | **0.739** | 0.731 | 0.731 | 0.728 | 0.725 | 0.565 | 0.582 | 0.574 | 0.547 | 0.552 |
| DINOv2 | 0.764 | 0.796 | 0.795 | 0.767 | 0.763 | 0.730 | 0.731 | 0.732 | 0.730 | 0.726 | 0.558 | 0.582 | 0.582 | 0.560 | 0.554 |
| DINOv3 | 0.789 | 0.813 | **0.817** | 0.780 | 0.777 | 0.731 | 0.734 | 0.734 | 0.729 | 0.728 | 0.577 | 0.597 | 0.600 | 0.569 | 0.566 |

*Table 11.* **Segmentation results of AJI and Dice with different backbones on the CPM17 dataset.**

| Backbones | AJI | | | | Dice | | | |
|---|---|---|---|---|---|---|---|---|
| | 64 | 128 | 256 | 512 | 64 | 128 | 256 | 512 |
| UNI | 0.641 | 0.651 | 0.639 | 0.594 | 0.805 | 0.812 | 0.802 | 0.769 |
| UNI2 | 0.645 | 0.652 | 0.642 | 0.596 | 0.803 | 0.815 | 0.802 | 0.762 |
| Virchow | 0.643 | 0.649 | 0.646 | 0.599 | 0.805 | 0.812 | 0.807 | 0.769 |
| Virchow2 | 0.642 | 0.640 | 0.635 | 0.592 | 0.794 | 0.806 | 0.787 | 0.768 |
| H-optimus-1 | 0.651 | **0.662** | 0.643 | 0.580 | 0.804 | 0.818 | 0.795 | 0.765 |
| ViT-l/16 | 0.649 | 0.654 | 0.642 | 0.583 | 0.809 | 0.810 | 0.804 | 0.759 |
| DINOv2 | 0.643 | 0.649 | 0.637 | 0.584 | 0.805 | 0.809 | 0.800 | 0.759 |
| DINOv3 | 0.648 | 0.657 | 0.637 | 0.595 | 0.805 | **0.821** | 0.799 | 0.766 |

*Table 12.* **Segmentation results of DQ, SQ, and PQ with different backbones on the CPM17 dataset.**

| Backbones | DQ | | | | SQ | | | | PQ | | | |
|---|---|---|---|---|---|---|---|---|---|---|---|---|
| | 64 | 128 | 256 | 512 | 64 | 128 | 256 | 512 | 64 | 128 | 256 | 512 |
| UNI | 0.777 | 0.787 | 0.787 | 0.739 | 0.768 | 0.770 | 0.771 | 0.763 | 0.597 | 0.606 | 0.606 | 0.564 |
| UNI2 | 0.771 | 0.786 | 0.780 | 0.739 | 0.763 | 0.768 | 0.770 | 0.763 | 0.588 | 0.604 | 0.601 | 0.564 |
| Virchow | 0.773 | 0.781 | 0.793 | 0.743 | 0.769 | 0.771 | 0.768 | 0.764 | 0.594 | 0.602 | 0.609 | 0.568 |
| Virchow2 | 0.767 | 0.780 | 0.783 | 0.738 | 0.768 | 0.769 | 0.770 | 0.768 | 0.589 | 0.600 | 0.602 | 0.567 |
| H-optimus-1 | 0.774 | **0.796** | 0.776 | 0.734 | 0.768 | **0.774** | 0.770 | 0.763 | 0.594 | **0.616** | 0.599 | 0.560 |
| ViT-l/16 | 0.784 | 0.789 | 0.787 | 0.725 | 0.767 | 0.769 | 0.772 | 0.761 | 0.601 | 0.607 | 0.608 | 0.552 |
| DINOv2 | 0.783 | 0.789 | 0.780 | 0.726 | 0.768 | 0.767 | 0.771 | 0.766 | 0.601 | 0.605 | 0.604 | 0.556 |
| DINOv3 | 0.788 | 0.794 | 0.789 | 0.747 | 0.768 | 0.770 | 0.772 | 0.766 | 0.605 | 0.611 | 0.609 | 0.572 |

*Table 13.* **Segmentation results of AJI and Dice with different backbones on the TNBC dataset.**

| Backbones | AJI | | | | Dice | | | |
|---|---|---|---|---|---|---|---|---|
| | 64 | 128 | 256 | 512 | 64 | 128 | 256 | 512 |
| UNI | 0.601 | 0.600 | 0.587 | 0.566 | 0.767 | 0.773 | 0.761 | 0.746 |
| UNI2 | 0.602 | 0.593 | 0.587 | 0.577 | 0.761 | 0.765 | 0.761 | 0.754 |
| Virchow | 0.595 | 0.605 | 0.593 | 0.568 | 0.767 | 0.775 | 0.766 | 0.747 |
| Virchow2 | 0.592 | 0.599 | 0.575 | 0.565 | 0.767 | 0.769 | 0.752 | 0.742 |
| H-optimus-1 | 0.603 | 0.600 | 0.576 | 0.574 | 0.768 | **0.780** | 0.753 | 0.748 |
| ViT-l/16 | **0.609** | 0.604 | 0.594 | 0.570 | 0.774 | 0.766 | 0.766 | 0.746 |
| DINOv2 | 0.606 | 0.596 | 0.581 | 0.567 | 0.772 | 0.766 | 0.755 | 0.747 |
| DINOv3 | 0.595 | 0.576 | 0.580 | 0.576 | 0.761 | 0.759 | 0.756 | 0.754 |

*Table 14.* **Segmentation results of DQ, SQ, and PQ with different backbones on the TNBC dataset.**

| Backbones | DQ | | | | SQ | | | | PQ | | | |
|---|---|---|---|---|---|---|---|---|---|---|---|---|
| | 64 | 128 | 256 | 512 | 64 | 128 | 256 | 512 | 64 | 128 | 256 | 512 |
| UNI | 0.783 | 0.788 | 0.782 | 0.766 | 0.785 | 0.788 | 0.787 | 0.781 | 0.615 | 0.621 | 0.615 | 0.598 |
| UNI2 | 0.780 | 0.790 | 0.787 | 0.767 | **0.789** | 0.787 | 0.786 | 0.782 | 0.615 | 0.622 | 0.619 | 0.600 |
| Virchow | 0.780 | **0.796** | 0.794 | 0.767 | 0.785 | 0.787 | 0.784 | 0.781 | 0.612 | **0.626** | 0.622 | 0.599 |
| Virchow2 | 0.792 | 0.795 | 0.776 | 0.763 | 0.786 | 0.785 | 0.786 | 0.784 | 0.623 | 0.624 | 0.610 | 0.598 |
| H-optimus-1 | 0.785 | 0.794 | 0.774 | 0.775 | 0.787 | 0.786 | 0.783 | 0.779 | 0.618 | 0.624 | 0.606 | 0.604 |
| ViT-l/16 | 0.794 | 0.793 | 0.788 | 0.768 | 0.782 | 0.785 | 0.785 | 0.781 | 0.621 | 0.623 | 0.619 | 0.600 |
| DINOv2 | 0.794 | **0.796** | 0.781 | 0.772 | 0.784 | 0.787 | 0.785 | 0.778 | 0.622 | **0.626** | 0.613 | 0.601 |
| DINOv3 | 0.784 | 0.783 | 0.771 | 0.775 | 0.788 | 0.785 | 0.785 | 0.781 | 0.618 | 0.615 | 0.605 | 0.605 |

*Table 15.* **Segmentation results of AJI and Dice with different backbones on the PanNuke dataset.**

| Backbones | AJI | | | Dice | | |
|---|---|---|---|---|---|---|
| | 64 | 128 | 256 | 64 | 128 | 256 |
| UNI | 0.571 | 0.578 | 0.572 | 0.757 | 0.763 | 0.756 |
| UNI2 | 0.574 | 0.580 | 0.571 | 0.760 | **0.766** | 0.759 |
| Virchow | 0.574 | 0.578 | 0.575 | 0.759 | 0.764 | 0.762 |
| Virchow2 | 0.563 | 0.569 | 0.562 | 0.751 | 0.756 | 0.752 |
| H-optimus-1 | 0.576 | 0.580 | 0.566 | 0.761 | **0.766** | 0.754 |
| ViT-l/16 | 0.577 | **0.581** | 0.568 | 0.762 | **0.766** | 0.756 |
| DINOv2 | 0.578 | 0.577 | 0.563 | 0.762 | 0.763 | 0.753 |
| DINOv3 | 0.571 | 0.571 | 0.573 | 0.758 | 0.758 | 0.760 |

*Table 16.* **Segmentation results of DQ, SQ, and PQ with different backbones on the PanNuke dataset.**

| Backbones | DQ | | | SQ | | | PQ | | |
|---|---|---|---|---|---|---|---|---|---|
| | 64 | 128 | 256 | 64 | 128 | 256 | 64 | 128 | 256 |
| UNI | 0.685 | 0.689 | 0.692 | 0.760 | 0.761 | 0.758 | 0.521 | 0.524 | 0.525 |
| UNI2 | 0.687 | 0.699 | 0.695 | 0.760 | 0.762 | 0.759 | 0.522 | 0.533 | 0.528 |
| Virchow | 0.690 | 0.699 | 0.696 | 0.760 | 0.760 | 0.759 | 0.524 | 0.531 | 0.528 |
| Virchow2 | 0.679 | 0.687 | 0.696 | 0.760 | 0.760 | 0.760 | 0.516 | 0.522 | 0.529 |
| H-optimus-1 | 0.687 | **0.700** | 0.688 | 0.761 | **0.763** | 0.757 | 0.523 | **0.534** | 0.521 |
| ViT-l/16 | 0.693 | **0.700** | 0.687 | 0.761 | 0.760 | 0.757 | 0.527 | 0.532 | 0.520 |
| DINOv2 | 0.695 | 0.697 | 0.684 | 0.761 | 0.761 | 0.757 | 0.529 | 0.530 | 0.518 |
| DINOv3 | 0.690 | 0.694 | 0.683 | 0.759 | 0.760 | 0.760 | 0.524 | 0.527 | 0.519 |

To quantify the difference between SAM and its domain-adapted variants, we replace SAM with MedSAM and SAM-Med2D (Cheng et al., 2023). As shown in Table 17, the results remain close to those obtained with the original SAM-b backbone on both MoNuSeg and CPM17 datasets. This suggests that our pipeline effectively leverages the segmentation capability of the original SAM, achieving performance comparable to that of domain-adapted variants.

*Table 17.* **Comparison of SAM and domain-adapted segmentation backbones on MoNuSeg and CPM17.** MedSAM and SAM-Med2D are evaluated using checkpoints built on SAM-b.

| Method | MoNuSeg | | CPM17 | |
|---|---|---|---|---|
| | AJI | Dice | AJI | Dice |
| MedSAM | 0.609 | 0.786 | 0.654 | 0.811 |
| SAM-Med2D | 0.606 | 0.782 | 0.651 | 0.809 |
| SAM-b | 0.615 | 0.787 | 0.649 | 0.803 |

### E.4. Clustering Configuration

To evaluate the influence of the clustering configuration, we replace the original $K$-means with $K$-means++ and GMM, and vary the number of prototype centers on MoNuSeg. As shown in Table 18, the AJI scores vary within a small range across different prototype numbers, indicating that the proposed method is not sensitive to the clustering strategy or the number of prototype centers. GMM yields slightly lower performance than $K$-means

*Table 18.* **Comparison of clustering configurations on MoNuSeg.**

| Method | 2 | 3 | 4 | 5 | 6 |
|---|---|---|---|---|---|
| K-means | 0.614 | 0.621 | 0.618 | 0.616 | 0.620 |
| K-means++ | 0.613 | 0.620 | 0.616 | 0.619 | 0.612 |
| GMM | 0.610 | 0.613 | 0.607 | 0.606 | 0.609 |

and $K$-means++, which is consistent with the intuition that soft assignments may produce less distinctive prototype centers.

### E.5. Point Quality and Generation

*Table 19.* **Comprehensive ablation results of point generation on MoNuSeg.** Partial denotes POT-Scan, while balanced refers to standard OT. When OT is not applied, similarity mapping is performed using a greedy strategy. The baseline corresponds to SAM auto-prompting.

| Color Prior Mask | Similarity Mapping | OT | AJI | PQ | DQ | SQ | Dice |
|---|---|---|---|---|---|---|---|
| ✗ | ✗ | ✗ | 0.061 | 0.262 | 0.384 | 0.752 | 0.353 |
| Otsu | ✗ | ✗ | 0.527 | 0.468 | 0.634 | 0.738 | 0.725 |
| Otsu | ✓ | ✗ | 0.532 | 0.468 | 0.636 | 0.736 | 0.728 |
| Otsu | ✓ | balanced | 0.545 | 0.474 | 0.647 | 0.732 | 0.742 |
| Otsu | ✓ | partial | 0.579 | 0.485 | 0.661 | 0.733 | 0.767 |
| High-confidence | ✓ | ✗ | 0.543 | 0.479 | 0.652 | 0.735 | 0.738 |
| High-confidence | ✓ | balanced | 0.567 | 0.510 | 0.686 | 0.743 | 0.759 |
| High-confidence | ✓ | partial | 0.619 | 0.577 | 0.787 | 0.733 | 0.795 |

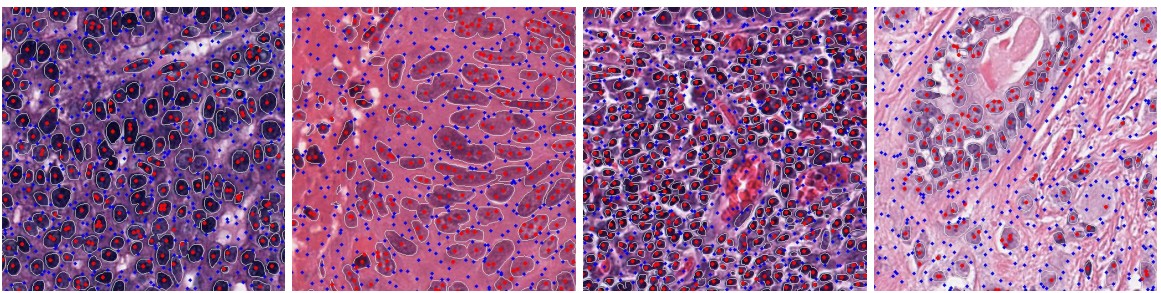

*Figure 21.* **Visualization of point prompts. (MoNuSeg)** Positive prompts (red dots) and negative prompts (blue squares) are overlaid on the images to illustrate the quality of prompt selection, with white contours denoting the ground-truth cell boundaries. Images are cropped for clarity of visualization.

We report the detailed quantitative metrics corresponding to the ablation results presented in Section 4.2 Figure 7 in Table 19. These results provide a comprehensive view of how each component contributes to the overall segmentation performance.

To evaluate the generated point prompts both quantitatively and qualitatively, we present the visualization of the point

*Table 20.* **Evaluation of the generated point prompt quality on different datasets.** The prompts exhibit high accuracy, with reliable positive-point coverage and robust background suppression from negative points.

| Datasets | TP | TN | FP | FN |
|---|---|---|---|---|
| MoNuSeg | 0.849 | 0.943 | 0.151 | 0.057 |
| CPM17 | 0.873 | 0.956 | 0.127 | 0.044 |
| TNBC | 0.818 | 0.982 | 0.182 | 0.018 |

spatial distribution overlaid with the images in Figure 21 and the direct accuracy of both positive and negative points across the three datasets in Table 20. Overall, both positive and negative prompts exhibit high precision, with negative points achieving particularly strong true negative rates by effectively avoiding non-cell areas. Positive points also demonstrate robust coverage since they reliably capture nuclei even in densely packed regions with small cells. For larger nuclei, multiple positive points are often assigned within a single object, further increasing the possibility of accurate segmentation. Only a minor fraction of very small or weakly stained nuclei may be underrepresented, reflecting inherent challenges in such pathological settings. This strong alignment supports the downstream instance prediction, as accurate point allocation provides a reliable foundation for mask generation.

### E.6. Prompt Sampling Strategy

We further ablate the prompt sampling strategy used for foreground and background prompt generation. For foreground prompts, we compare random sampling, distance-transform (DT) peak sampling, and watershed-based sampling. For background prompts, we compare random sampling, grid sampling, and grid-based uniform sampling. As shown in Table 21, watershed-based foreground sampling achieves the best performance, suggesting that it localizes instance centers more reliably than random or DT peak sampling. For background prompts, grid-based uniform sampling performs slightly better than the alternatives, due to its more even exclusion of non-target regions.

*Table 21.* **Ablation study on prompt sampling strategies on MoNuSeg.**

| Sampling | Type | AJI | Dice |
|---|---|---|---|
| Random | FG | 0.594 | 0.771 |
| DT peak | FG | 0.608 | 0.783 |
| Watershed | FG | 0.621 | 0.795 |
| Random | BG | 0.606 | 0.784 |
| Grid | BG | 0.613 | 0.788 |
| Grid-based uniform | BG | 0.621 | 0.795 |

*Table 22.* **Detailed ablation results of post-processing strategy on MoNuSeg.** CP denotes containment penalty.

| CP | NMS | AJI | PQ | DQ | SQ | Dice |
|---|---|---|---|---|---|---|
| ✗ | hard | 0.553 | 0.523 | 0.712 | 0.735 | 0.743 |
| ✗ | soft | 0.573 | 0.529 | 0.726 | 0.729 | 0.765 |
| ✗ | hard+soft | 0.565 | 0.534 | 0.719 | 0.743 | 0.757 |
| ✓ | hard | 0.585 | 0.573 | 0.785 | 0.730 | 0.764 |
| ✓ | soft | 0.620 | 0.593 | 0.809 | 0.733 | 0.789 |
| ✓ | hard+soft | 0.605 | 0.576 | 0.788 | 0.731 | 0.778 |

### E.7. Soft NMS

In this section, we present a comprehensive evaluation of the proposed post-processing strategy. We report detailed performance metrics and ablation studies on the score and decay functions of soft NMS to validate the method. We further include visualizations and pseudo-code to illustrate the procedure.

As a complement to Figure 7b, Table 22 reports the detailed ablation results of the NMS strategy across all five evaluation metrics.

To assess the necessity of the combined score strategy, we compare H-channel response alone, SAM confidence score alone, and their combination, with the raw predictions as the baseline. As shown in Table 23, applying soft NMS improves the overall segmentation performance compared with directly aggregating predictions. Both the H-channel response and the SAM score alone further enhance performance. This highlights the importance of incorporating image-derived cues

*Table 23.* **Ablation of score strategies for soft NMS on MoNuSeg.** Results highlight different focuses between H-channel response and SAM confidence. The combined approach provides the most balanced improvement in post-processing.

| Strategy | AJI | PQ | DQ | SQ | Dice |
|---|---|---|---|---|---|
| ✗ | 0.546 | 0.364 | 0.495 | 0.735 | 0.747 |
| $S'_H$ | 0.591 | 0.581 | 0.793 | 0.733 | 0.784 |
| $S_{SAM}$ | 0.587 | 0.583 | 0.792 | 0.732 | 0.781 |
| $S'_H + S_{SAM}$ | 0.619 | 0.593 | 0.809 | 0.734 | 0.792 |

*Table 24.* **Ablation of decay functions for soft NMS on MoNuSeg.** All soft NMS variants outperform hard NMS. Exponential decay achieves the best overall performance by more effectively penalizing heavily overlapping instances while preserving valid detections.

| Functions | AJI | PQ | DQ | SQ | Dice |
|---|---|---|---|---|---|
| hard | 0.587 | 0.549 | 0.761 | 0.721 | 0.764 |
| linear | 0.601 | 0.564 | 0.774 | 0.729 | 0.778 |
| polynomial | 0.599 | 0.568 | 0.778 | 0.731 | 0.780 |
| exponential | 0.614 | 0.577 | 0.787 | 0.733 | 0.787 |

when refining instance predictions. The H-channel emphasizes heavily stained regions but tends to keep darker tissue areas. In contrast, the SAM score favors nuclei with clear boundaries, preserving large nuclear regions but failing to separate overlapping cells. The combined strategy effectively balances these complementary strengths and achieves the best overall performance.

We further investigate the effect of different score decay functions in soft NMS, including linear, polynomial, and exponential. Hard NMS, as a conventional suppression scheme, is employed as the baseline. As summarized in Table 24, all variants of Soft-NMS consistently outperform hard NMS across multiple metrics, underscoring the advantage of retaining slight overlapping predictions with gradually decayed scores rather than discarding them. This flexible suppression handles dense cell regions better. Among the tested functions, the exponential decay yields the best overall performance. This can be attributed to the fact that exponential decay provides a sharper penalty to highly overlapping instances while still preserving moderately overlapping candidates. It strikes a better balance between suppressing redundant de-

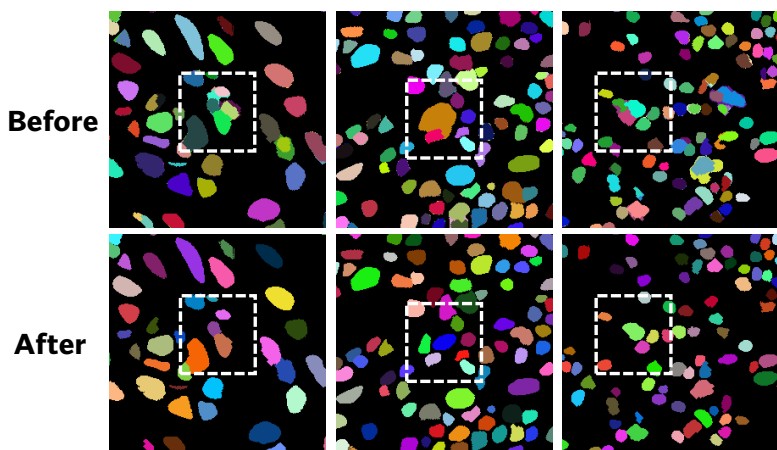

*Figure 22.* **Visualization of soft NMS post-processing.** The containment-aware variant effectively suppresses large predicted masks that contain smaller ones, leading to cleaner and more precise outputs. The highlighted regions show distinct differences.

tections and retaining true positives in the post-processing procedure. The effectiveness of soft NMS with the exponential decay function is illustrated in Figure 22.

Lastly, we provide the comprehensive pseudo-code of the containment-aware soft NMS in Alg.(3).

## F. Additional Sensitivity Analysis

### F.1. SAM Effect

SAM is highly sensitive to the form of inputs. Naïve application often leads to suboptimal performance on pathology images due to densely packed structures, heterogeneous staining, and subtle nuclear boundaries. The strategies illustrated in the Figure 23 are designed to mitigate these challenges.

**Image size.** As illustrated in Figure 23a, scaling controls the receptive field over which SAM attends. Without scaling,

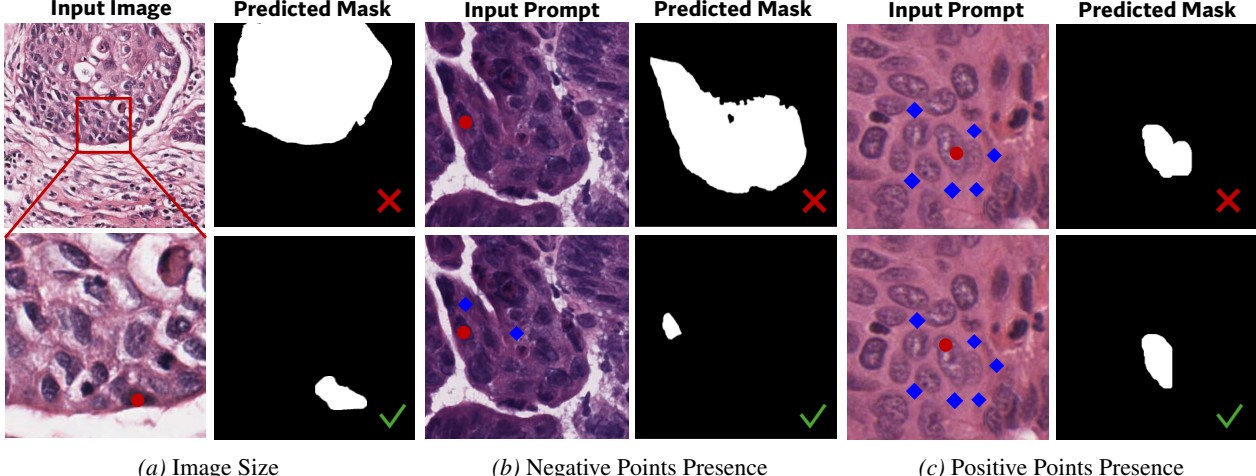

| Input Image | Predicted Mask | Input Prompt | Predicted Mask | Input Prompt | Predicted Mask |

*(a)* Image Size       *(b)* Negative Points Presence       *(c)* Positive Points Presence

*Figure 23.* **Illustration of SAM effect.** (a) Appropriate image cropping enables SAM to focus on local nuclei instead of broad tissue regions. (b) Incorporating negative prompts suppresses background responses for precise nuclear delineation, which is especially valuable in images where stained tissue and nuclei exhibit similar colors. (c) The placement of positive prompts influences the separation of overlapping nuclei. Red dots denote positive prompts and blue squares denote negative prompts.

SAM can respond to large tissue regions, ignoring fine-grained nuclear detail. By splitting the image into small patches, the model is guided to treat individual nuclei as primary segmentation targets.

**Negative prompts.** In histology images, nuclei are frequently embedded within complex backgrounds where clear boundaries are lacking. As shown in Figure 23b, negative prompts act as explicit counterexamples, guiding SAM to disregard surrounding tissue that mimics nuclear appearance. This strategy mitigates over-segmentation and enhances boundary precision.

**Positive prompt placement.** Overlapping or touching nuclei pose a significant challenge, as SAM may merge them into a single mask in Figure 23c. Carefully placing positive prompts within each nucleus provides local anchors that enforce instance-level separation, enabling SAM to segment individual objects rather than clusters.

Together, these strategies systematically adapt SAM's general-purpose design to the demands of nuclear segmentation. They highlight that effective use of SAM in computational pathology depends not only on model capacity but also on thoughtful construction of input prompts that reflect domain-specific image structure.

## G. Discussion and Future Directions

### G.1. Beyond H&E

SPROUT is fundamentally designed for H&E pathology images, where the stain-derived optical density provides a biochemical prior for constructing high-confidence foreground and background masks to guide prototype formation. Consequently, all main experiments in the paper focus on H&E nuclear segmentation.

---

**Algorithm 3** Containment-aware Soft NMS

1: **Inputs:** Masks $\{M_i\}_1^n$, scores $\{S_i\}_1^n$
2: Exponential decay $f(x) = \exp(-x^2/\sigma)$ with $\sigma > 0$, penalty scale $\epsilon > 0$, score threshold $\tau$.
3:
4: $\mathcal{B} \leftarrow \emptyset$
5: $\mathcal{D} \leftarrow \{(M_i, S_i)\}_{i=1}^n$      ▷ working set of detections
6:
7: **for** each $(M_i, S_i) \in \mathcal{D}$ **do**
8:     **if** $N_{\text{contained}}(M_i) > 1$ **then**
9:        $S_i \leftarrow S_i \cdot \left(1 - \tanh(\epsilon \cdot N_{\text{contained}}(M_i))\right)$
10:     **end if**
11: **end for**
12:
13: **while** $\mathcal{D} \neq \emptyset$ **do**
14:     $(M_{\max}, S_{\max}) \leftarrow \arg\max_{(M,S) \in \mathcal{D}} S$
15:     $\mathcal{B} \leftarrow \mathcal{B} \cup \{M_{\max}\}$
16:     $\mathcal{D} \leftarrow \mathcal{D} \setminus \{(M_{\max}, S_{\max})\}$
17:     **for** each $(M_i, S_i) \in \mathcal{D}$ **do**
18:        $b_{\max} \leftarrow \text{bbox}(M_{\max}), \quad b_i \leftarrow \text{bbox}(M_i)$
19:        $u \leftarrow \text{IoU}(b_{\max}, b_i)$
20:        $S_i \leftarrow S_i \cdot f(u)$
21:        **if** $S_i < \tau$ **then**
22:          $\mathcal{D} \leftarrow \mathcal{D} \setminus \{(M_i, S_i)\}$
23:        **end if**
24:     **end for**
25: **end while**
26:
27: **return** $\mathcal{B}$

---

To explore the extensibility of the proposed framework, we conducted a qualitative experiment on fluorescent nuclei images from the DSB2018 dataset (Caicedo et al., 2019). For this modality, we replace the stain-prior module with a fluorescence-intensity prior, where bright emission regions correspond to nuclear structures. Since no backbone is specifically trained for fluorescent microscopy images, we adopt natural-image pretrained backbones instead to extract features for prototype construction and POT-Scan refinement.

Despite the absence of color semantics, SPROUT's prototype clustering and POT-Scan refinement remain effective. As shown in Figure 24, the resulting segmentation closely matches the provided ground truth. These results suggest that the core mechanism of SPROUT is not bound to H&E-specific color cues, but can operate as long as the modality provides reliable signals to distinguish foreground nuclei from background tissue.

This observation highlights a promising direction for future work. SPROUT can potentially be extended to other imaging modalities by developing modality-specific cues, such as color characteristics, intensity patterns, or structural contrast, to generate high-confidence foreground and background indicators for self-reference and feature allocation. We hope that SPROUT will motivate further exploration on training-free strategies for nuclear segmentation and encourage the community to investigate a broader range of cellular imaging modalities beyond traditional H&E pathology.

### G.2. Practical Q&A

**Question 1:   Does SPROUT generalize beyond H&E-stained images?**

**Answer 1:**   SPROUT is primarily evaluated on H&E-stained images, which represent the most common and clinically relevant pathology modality. However, the framework is not intrinsically tied to H&E staining. SPROUT only assumes the availability of reference foreground and background color cues for reliable separation and prototype construction, after which the same pipeline can be applied. How to automatically identify such cues across diverse pathology modalities will be an interesting direction for future work.

**Question 2:   How does SPROUT scale to whole-slide images (WSIs) and large datasets?**

**Answer 2:**   SPROUT scales naturally to whole-slide images through its patch-based and fully local design. WSI processing does not introduce a new methodological setting, but only increases the number of patches to be processed. In practice, segmentation is applied only to regions of interest, allowing large blank areas to be skipped. Since SPROUT operates without slide-level supervision or learned parameters, the resulting runtime remains practically manageable under standard pathology workflows and primarily reflects engineering throughput rather than algorithmic limitations.

**Question 3:   What is the practical use case of SPROUT in real-world pathology workflows?**

**Answer 3:**   SPROUT is particularly well-suited for real-world pathology workflows where annotations and training are impractical or undesirable. SPROUT can be directly applied to private or institution-specific datasets where data sharing and annotation are restricted. It enables rapid, large-scale generation of nuclear instance segmentation results. In addition, SPROUT can serve as a strong initialization for downstream tasks, such as morphology analysis or weak supervision, and as a reliable baseline to assess data characteristics before committing resources to annotation or model training.

**Question 4:   When might SPROUT fail?**

*Figure 24.* **Visualization of fluorescent nuclei segmentation on the DSB2018 dataset.** SPROUT is implemented with a ViT backbone for feature extraction, and SAM-Large for instance prediction. The segmentation results closely match the ground truth, demonstrating the conceptual extensibility of the proposed framework beyond H&E.

**Answer 4:** SPROUT may underperform in the presence of severe staining corruption, atypical color distributions, extremely dense nuclear clusters, or background structures with a nuclear-like appearance. Since it is training-free and appearance-driven, its performance depends on the reliability of visual cues.

Qualitatively, the main bottleneck lies in self-reference construction, where severe stain artifacts or heavily stained backgrounds may distort optical-density cues and produce unreliable references. POT-Scan can further propagate biased prototypes into diffuse or misplaced foreground maps, whereas SAM prompting is comparatively stable once reasonable prompts are generated. Therefore, failures mainly arise when corrupted or misleading visual cues affect the early reference and prototype construction stages.

**Question 5: How should the backbone, resolution, SAM weights, and hyperparameters be chosen in practice?**

**Answer 5:** SPROUT is not highly sensitive to the choice of backbone, resolution, or SAM weights. We observe that both natural-image–trained and pathology-trained backbones yield comparable performance once self-reference prompting is applied. For resolution, we recommend selecting a patch size whose effective receptive field is roughly aligned with the typical nuclear scale. SAM variants can be selected based on available computational resources, with larger models providing more accurate segmentation and smaller ones offering efficient inference with modest performance degradation. Hyperparameters are chosen to avoid extreme behaviors rather than dataset-specific tuning, for example, by ensuring sufficient reference masks, multiple prototype centers, the presence of negative prompts, and moderate patch overlap. We find that such non-extreme choices give stable and comparable performance in practice.

