# OpenReview forum: "Supervise Less, See More: Training-free Nuclear Instance Segmentation with Prototype-Guided Prompting"
_ICML.cc/2026/Conference — ICML 2026 regular_

### Official Review · Reviewer_Zp44 · 2026-02-19

**Soundness:** 4
**Presentation:** 4
**Significance:** 3
**Originality:** 3
**Overall Recommendation:** 5
**Confidence:** 4

**Summary:**

This paper proposes SPROUT, a fully training free and annotation free framework for nuclei instance segmentation. It first exploits hematoxylin and eosin optical density priors to extract high confidence foreground and background regions and build feature prototypes. A progressive partial optimal transport algorithm with a slack column then aligns image features to these prototypes while filtering noisy regions as background. The aligned features are converted into point prompts and fed into a frozen Segment Anything Model, followed by a containment aware soft non maximum suppression to refine overlaps. Without any fine tuning, SPROUT achieves competitive results against several supervised and weakly supervised methods on MoNuSeg, CPM17, TNBC, and PanNuke.

**Compliance With Llm Reviewing Policy:**

Affirmed.

**Key Questions For Authors:**

1. While Appendix G reports an extension to fluorescence images, multiplex immunohistochemistry often lacks a stable biochemical foreground background contrast that can be reliably captured by optical density thresholding. Do the authors have a concrete plan to replace the current optical density based priors with other forms of zero shot priors?

2. The containment aware soft NMS tends to penalize large masks that contain multiple smaller regions. However, in certain tumor subtypes, multinucleated giant cells can exhibit genuine biological structures where a large nucleus overlaps with or contains multiple smaller nuclear regions. Could this penalty mechanism lead to systematic over segmentation in such cases, and have the authors observed any failure modes or considered safeguards to avoid suppressing biologically valid containment patterns?

**Limitations:**

Yes

**Strengths And Weaknesses:**

Strengths:
In the field of computational pathology that heavily depends on dense pixel level annotations, proposing a completely training free framework is both bold and highly inspiring. The paper creatively bridges classical biochemical priors from hematoxylin and eosin channel separation with modern vision foundation models through optimal transport theory, resulting in a conceptually novel and elegant design.

The POT-Scan module is not a simple heuristic combination. In Appendix A, the authors provide a rigorous mathematical derivation from standard optimal transport to partial optimal transport with a slack column, together with proofs of convergence and equivalence. This formal grounding offers strong justification for the model’s noise robustness.

Beyond validating the contribution of core components, the authors analyze different feature backbones, compare natural image and pathology specific encoders, examine various SAM model scales from Tiny to Large, study hyperparameter sensitivity, and evaluate different post processing decay functions. Such comprehensive experimentation substantially strengthens the credibility of the conclusions.

Weaknesses:
The first stage of SPROUT relies heavily on the physical properties of hematoxylin and eosin staining, where hematoxylin highlights nuclei and eosin stains cytoplasm. Although Appendix G.1 presents preliminary results on fluorescence images, extending the framework seamlessly to more diverse modalities such as immunohistochemistry or multiplex immunofluorescence may be challenging, as intensity based single channel priors could become less reliable.

While training cost is eliminated, each image still requires feature clustering and the execution of a Sinkhorn based optimal transport solver. According to the runtime analysis in Appendix D.2, although parallelization is possible, the wall clock time per image remains significantly higher than a single forward pass of a conventional end to end U-Net model.

---

> ### Author Rebuttal · Authors · 2026-03-31
>
> Dear Reviewer `Zp44`,
>
> We thank the reviewer for the careful reading of our paper and for the detailed and insightful comments! We have carefully revised the manuscript to address each of the points raised.
>
> ---
>
> > W1&Q1: How can SPROUT be extended to modalities like multiplex immunohistochemistry, where stain priors are unreliable, and what alternative zero-shot priors could replace them?
>
> A: Thank you for the question! Our framework does not fundamentally rely on OD-based priors. The key requirement is **finding a zero-shot prior that identifies high-confidence foreground and background regions for prototype construction**. In Appendix G, we demonstrate this modularity by replacing the H&E-based prior with a fluorescence-intensity prior, while keeping the rest unchanged.
>
> For multiplex IHC, we would similarly replace the stain prior with a marker-channel-based prior, such as nuclear-marker intensity or channel-wise probability maps to identify high-confidence regions. The feature extractor can be adapted to the modality as well, either by adopting a domain-specific encoder or a natural-image encoder. In this way, the prior module is modality-dependent, but the downstream framework can be reused.
>
> > W2: While training cost is eliminated, each image still requires feature clustering and the execution of a Sinkhorn-based OT solver. According to the runtime analysis in Appendix D.2, although parallelization is possible, the wall clock time per image remains significantly higher than a single forward pass of a conventional end-to-end U-Net model.
>
> A: Thank you for the comment! We agree that SPROUT is slower at inference than a single-pass U-Net. We view it as a **trade-off between higher test-time latency and lower target-dataset setup cost**. To better contextualize it, we compare SPROUT on CPM17 with three representative paradigms: a conventional end-to-end supervised model (U-Net), a SAM-based domain-adapted model (MedSAM), and an annotation-free method (COIN). Both U-Net and MedSAM rely on expensive annotation, while training COIN requires intensive SAM prediction. As a result, SPROUT has moderately higher inference time, but delivers better performance while keeping the overall cost low. We hope this addresses your concerns, and we welcome any further discussion.
>
> | Method | Extra train? | Ann.? | New data retrain? | s/img | AJI ↑ |
> | --- | :---: | :---: | :---: | --- | --- |
> | U-Net | ✓ | ✓ | ✓ | 0.294 | 0.554 |
> | MedSAM | ✓ | ✓ | ✗ | 7.242 | 0.648 |
> | COIN | ✓ | ✗ | ✓ | 4.311 | 0.572 |
> | SPROUT | ✗ | ✗ | ✗ | 9.452 | 0.662 |
>
> > Q2: Could containment-aware soft NMS over-segment biologically valid containment patterns, such as multinucleated giant cells? Have the authors observed any failure modes or considered safeguards to avoid suppressing such patterns?
>
> A: Good Point! We agree that an overly strong containment penalty could risk over-segmenting multinucleated giant cells or highly atypical tumor morphologies. We would like to clarify two points.
>
> - The datasets used in our paper (MoNuSeg, PanNuke, CPM17, and TNBC) are dominated by conventional nuclear morphologies, and we did not observe this as a systematic failure mode in our benchmarks.
> - Our containment-aware refinement uses a **soft**, rather than hard, penalty to down-rank large enclosing masks likely caused by SAM merging adjacent nuclei, while still jointly considering SAM confidence and stain-prior response. Therefore, it suppresses obvious over-merged predictions without explicitly rejecting biologically plausible containment patterns.
>
> A practical safeguard would be to reduce or disable the containment penalty when such cell morphologies are expected, since the penalty strength is adjustable. Thanks for bringing it up, and we hope our response addresses your concerns.

---

> > ### Author Rebuttal · Reviewer_Zp44 · 2026-04-01
> >
> > The author has addressed all of my concerns.

---

> > > ### Author Response · Authors · 2026-04-01
> > >
> > > Dear Reviewer `Zp44`,
> > >
> > > Thank you very much for your response and for taking the time to read our rebuttal. We sincerely appreciate your thoughtful and constructive comments throughout the review process. Your questions helped us improve the clarity and completeness of the paper.
> > >
> > > We appreciate your careful consideration of our work. Thank you very much again!
> > >
> > > Sincerely,
> > >
> > > The Authors of Submission 19778

---

### Official Review · Reviewer_yDSH · 2026-02-26

**Soundness:** 3
**Presentation:** 3
**Significance:** 3
**Originality:** 2
**Overall Recommendation:** 4
**Confidence:** 4

**Summary:**

This paper proposes a training-free pipeline for nuclear instance segmentation in H&E pathology images using prototype-guided prompting for Segment Anything (SAM). The method first constructs high-confidence foreground/background regions from stain deconvolution as a self-reference, then builds foreground/background prototypes and aligns whole-image features to these prototypes via a progressive partial optimal transport procedure (POT-Scan). The resulting activation maps are used to automatically generate positive/negative point prompts for SAM, and a containment-aware post-processing step reduces merged-instance errors in dense regions. The approach is evaluated on multiple nuclear segmentation benchmarks with standard instance metrics and includes ablations on key components and a runtime analysis.

**Compliance With Llm Reviewing Policy:**

Affirmed.

**Final Justification:**

I appreciate the authors’ thorough rebuttal. The additional analyses on stain robustness, prompt-sampling sensitivity, failure modes, implementation details, and lightweight adaptation baselines directly address my main questions and improve my confidence in the method’s robustness, reproducibility, and practical positioning. Overall, the rebuttal reinforces my original positive assessment rather than substantially changing it. I still view the paper’s main limitation as moderate conceptual novelty and somewhat specialized scope, but the work is technically solid, clearly presented, and practically useful for training-free nucleus instance segmentation. Therefore, my final recommendation remains positive and unchanged.

**Key Questions For Authors:**

+ How does performance change under stain normalization choices or synthetic stain perturbations (color jitter / OD scaling), especially since stain-based self-reference is a core step? A positive robustness result would increase my confidence in real-world applicability.

+ How sensitive are results to (i) transported mass schedule, (ii) number of prototypes, and (iii) the prompt sampling rules? A small sensitivity grid would clarify whether improvements are stable or require careful tuning.

+ Since the paper is training-free, could you include a budget-matched comparison to “lightweight adaptation” baselines to better contextualize when training-free is the right trade-off? This could affect my significance assessment.

+ Can you provide a more structured breakdown of failure modes and which stage fails (self-reference vs OT alignment vs SAM prompting)? This would improve soundness/presentation.

**Limitations:**

Yes

**Strengths And Weaknesses:**

Soundness.
+ Strengths: The pipeline is technically coherent and matches the constraints of the task: it leverages stain priors to obtain reliable self-reference regions, uses partial OT to avoid forcing ambiguous regions into prototypes, and produces SAM prompts in a fully automatic way. The paper supports its main claims with multi-dataset experiments, standard instance-segmentation metrics, and targeted ablations.


+ Weaknesses: Several parts rely on domain-specific heuristics and multi-stage design choices. While ablations help, the method’s robustness to stain/scanner variability and failure modes could be characterized more directly to fully justify reliability in real deployments.


Presentation.
+ Strengths: The overall narrative is easy to follow and the method is broken into clear stages (self-reference → prototype/OT → prompting → post-processing). Figures and ablations make it relatively straightforward to understand what each component contributes.


+ Weaknesses: Some implementation details that materially affect reproducibility would benefit from being more “front-and-center”. These details are present in parts/appendix, but they are important enough that a compact configuration table would improve clarity.


Significance.
+ Strengths: Training-free, label-free nuclear instance segmentation is a practically relevant setting in computational pathology, and a robust prompt-generation mechanism can make foundation models like SAM more usable without expensive re-training. The approach could be adopted as a strong baseline or preprocessing step in downstream pathology pipelines.


+ Weaknesses: The impact is somewhat specialized to pathology nuclei, and it is not fully established how broadly the prototype-guided prompting approach transfers to other dense-instance domains or other staining protocols without adaptation.


Originality.
+ Strengths: The paper offers a thoughtful combination of ingredients to solve a hard practical problem, with POT-Scan (progressive partial OT) as the most distinctive component in how it stabilizes prototype alignment for prompt generation.

+ Weaknesses: Many building blocks are established. The novelty is mainly in assembling them into a training-free, pathology-tailored prompting pipeline rather than introducing a broadly new learning principle.


Overall, I lean weak accept: the method is technically solid and practically useful, even if the conceptual novelty is moderate.

---

> ### Author Rebuttal · Authors · 2026-03-31
>
> Dear Reviewer `yDSH`,
>
> We thank the reviewer for the careful reading and for the detailed and insightful comments! We have carefully revised the manuscript to address each point raised.
>
> ---
>
> >W1&Q1: How robust is SPROUT to stain normalization choices or synthetic color/OD perturbations?
>
> A: Thank you for the comment! We have added robustness analyses under **moderate stain variation**.
>
> For **stain normalization**, we compared SPROUT with Macenko and Vahadane and observed only small differences. For **synthetic stain perturbations,** color jitter causes minor drops, while OD scaling is more challenging but does not cause collapse.
>
> | Setting | AJI | Dice |
> | --- | --- | --- |
> | Macenko | 0.619 | 0.799 |
> | Vahadane | 0.614 | 0.793 |
> | BCS ± 0.1 | 0.617 | 0.792 |
> | BCS ± 0.2 | 0.611 | 0.790 |
> | H ± 0.02 | 0.616 | 0.791 |
> | H ± 0.05 | 0.612 | 0.789 |
> | BCS ± 0.1, H ± 0.02 | 0.618 | 0.792 |
> | BCS ± 0.2, H ± 0.05 | 0.612 | 0.790 |
> | H×0.8 | 0.604 | 0.782 |
> | E×0.8 | 0.607 | 0.783 |
> | H×1.2 | 0.608 | 0.784 |
> | E×1.2 | 0.609 | 0.785 |
> | H×0.8, E×1.2 | 0.602 | 0.784 |
> | H×1.2, E×0.8 | 0.606 | 0.783 |
> | SPROUT | 0.621 | 0.795 |
>
> (B: Brightness, C: Contrast, S: Saturation, H: Hue)
>
> > W1&Q4: Can you provide a structured breakdown of failure modes?
>
> A: Thank you for the question! Qualitatively,
>
> - **Self-reference construction** is the main bottleneck, since severe stain artifacts or a heavily stained background can distort optical-density cues and produce unreliable references.
> - **POT-Scan** is a secondary failure source that may propagate biased prototypes into diffuse or misplaced foreground maps.
> - **SAM prompting** is comparatively stable once the prompts are reasonable.
>
> > W2: Important implementation details should be more front-and-center.
>
> A: Thank you for the comment! The default configuration is summarized below.
>
> | Stage | Setting |
> | --- | --- |
> | Self-reference | Mask ratio: 0.6 |
> | Feat. extraction | UNI2;  patch 128 / overlap 0.5 |
> | Proto. construction | K-means; K = 3 |
> | POT-Scan | Init. mass: 0.6; stride: 0.05 |
> | Prompting | Pos: watershed; Neg: grid-uniform; nearest 3 |
> | SAM inference | SAM-Large; patch 512 / overlap 0.5 |
> | Post-proc. | Soft-NMS; exp. decay; H response + SAM conf. |
>
> > W3: Transfer to other dense-instance domains or stains is not fully established.
>
> A: Thank you for the comment! The framework is not fundamentally tied to H&E. It only requires a zero-shot modality-specific for rough fg/bg identification. In Appendix G, we replace the H&E prior with a fluorescence-intensity prior, while keeping the rest unchanged.
>
> > W4: The novelty is mainly in assembling established blocks.
>
> A: Thank you for the comment! Our contributions are:
>
> - A **training-free, annotation-free inference-time adaptation framework** that formulates a new problem: given an unlabeled target image and a frozen foundation model, how can reliable instance-level prompts be constructed without supervision or parameter updates?
> - A self-reference mechanism that **converts domain priors** into slide-specific fg/bg prototypes without curated references.
> - POT-Scan performs **progressive, distribution-level feature–prototype assignment** that balances coverage and suppresses noise for prompt generation.
>
> > Q2: How sensitive are results to (i) transported mass schedule, (ii) number of prototypes, and (iii) the prompt sampling rules?
>
> A: Thank you for the questions! For (i) transported mass schedule and (ii) number of prototypes, Section 4.3 shows that varying the initial transported mass (0.5–0.9), stride (0.02–0.1), and K-means centers (1–7) yields stable performance.
>
> For (iii) prompt sampling rules, we compared watershed, random, and distance-transform (DT) peak sampling for **foreground** prompts, and random, grid, and grid-based uniform sampling for background prompts. Grid-based uniform sampling performs slightly better with a more even exclusion of non-target regions, while watershed outperforms others because it localizes instance centers more reliably.
>
> | Sampling | Type | AJI | Dice |
> | --- | --- | --- | --- |
> | Random | FG | 0.594 | 0.771 |
> | DT peak | FG | 0.608 | 0.783 |
> | Watershed | FG | 0.621 | 0.795 |
> | Random | BG | 0.606 | 0.784 |
> | Grid | BG | 0.613 | 0.788 |
> | Grid-based uniform | BG | 0.621 | 0.795 |
>
> > Q3: Could you include a budget-matched comparison to lightweight adaptation baselines?
>
> A: Thank you for the comment! We contextualize the trade-off in a low-resource budget. With 10 labeled images, vanilla SAM performs poorly because auto-prompting fails to localize nuclei, while a lightweight learned prompter [1] improves substantially but still requires labeled data and training.
>
> | Method | Labels | Train | Infer. (s) | AJI |
> | --- | --- | --- | --- | --- |
> | Vanilla SAM | 0 | 0 | 5.878 | 0.135 |
> | Prompter+SAM | 10 | ~10 min | 6.743 | 0.518 |
> | SPROUT | 0 | 0 | 9.452 | 0.662 |
>
> [1] Z. Shui, et al. “Unleashing the Power of Prompt-driven Nucleus Instance Segmentation.” ECCV 2024

---

> > ### Author Rebuttal · Reviewer_yDSH · 2026-04-04
> >
> > Thank you for the detailed rebuttal. The additional robustness analysis under stain normalization and synthetic perturbations, the structured discussion of failure modes, the summarized default configuration, and the extra comparisons on prompt sampling and lightweight adaptation all help address my main questions. These additions improve my confidence in the method’s robustness, reproducibility, and practical positioning. While broader transfer beyond H&E remains a natural direction for future work, I view that as a scope limitation rather than a blocking concern for this submission. Overall, the rebuttal adequately addresses my concerns, and my overall assessment remains positive.

---

> > > ### Author Response · Authors · 2026-04-04
> > >
> > > Dear Reviewer `yDSH`,
> > >
> > > We sincerely thank you for your careful follow-up and continued positive assessment of our work. We are encouraged that the additional analyses and experiments effectively addressed your questions and strengthened your confidence in our method. Your constructive feedback throughout the review process has been invaluable in refining both the practical positioning and the presentation of our paper. Thank you again for your generous time, thoughtful comments, and encouraging support.
> > >
> > > Best regards,
> > >
> > > The Authors of Submission 19778

---

### Official Review · Reviewer_15wc · 2026-03-06

**Soundness:** 3
**Presentation:** 4
**Significance:** 2
**Originality:** 2
**Overall Recommendation:** 4
**Confidence:** 2

**Summary:**

The paper proposes SPROUT, a fully training-free and annotation-free framework for nuclear instance segmentation in H&E histopathology images. SPROUT bypasses the need for dense annotations or computationally expensive fine-tuning of foundation models. It achieves this through a three-step pipeline: extracting self-reference prototypes using biochemical stain priors; employing Partial Optimal Transport to align dense patch features to these prototypes, generating robust positive and negative point prompts ; and prompting the Segment Anything Model followed by a containment-aware Soft-NMS post-processing step.

**Compliance With Llm Reviewing Policy:**

Affirmed.

**Final Justification:**

The rebuttal has saticfactorily addressed my main concerns.

**Key Questions For Authors:**

- Could the authors explicitly articulate the fundamental machine learning contribution of this work that extends beyond the engineering aggregation of existing tools (e.g. Otsu, K-means, POT, SAM)?
- If the H&E color prior is subject to severe domain shifts, staining artifacts, or tissue fading, the initial high-confidence masks will be corrupted. The prototypes are derived from simple Otsu thresholding on optical density maps, which may no longer be effective.
- Please explain the rationale behind the methodological inconsistency in prompt generation. If POT is critical for accurately identifying positive prompts, why is a simple uniform sampling heuristic sufficient for negative prompts.

**Limitations:**

yes

**Strengths And Weaknesses:**

Strengths :
The pipeline is logically sound. Leveraging intrinsic H&E stain priors to establish image-specific self-reference prototypes is a clever bypass to the traditional reference-image selection problem. Furthermore, the formulation of POT to explicitly handle ambiguous features by consigning them to a slack column is mathematically rigorous and highly appropriate for dense nuclei segmentation.

Weaknesses:
- While the integration of OT with foundation models is practically effective, the core algorithmic novelty from a strictly machine learning perspective is marginal. The formulation of POT via a slack column heavily relies on existing mathematical frameworks. The primary contribution is essentially an application-specific engineering pipeline.
- The methodological elegance of the POT-Scan is severely undermined by its heavy reliance on naive heuristics for initialization. But the prototypes are derived from simple Otsu thresholding applied to optical density maps based on standard H&E staining priors.

---

> ### Author Rebuttal · Authors · 2026-03-31
>
> Dear Reviewer `15wc`,
>
> Thank you for your thoughtful feedback! We have carefully revised the manuscript to address each of the points raised.
>
> ---
>
> > W1&Q1: What is the fundamental machine learning contribution of this work beyond integrating existing components?
>
> A: Thank you for the question! Our core ML contribution is the formulation of a new problem and a principled solution: *given an unlabeled target image and a frozen foundation model, how can we construct reliable instance-level prompts without supervision or parameter updates?* To our knowledge, this **inference-time unsupervised adaptation** setting has not been formally addressed for dense instance segmentation. Specifically:
>
> - The self-reference mechanism **converts domain priors** into slide-specific foreground and background prototypes without curated references.
> - POT-Scan performs **progressive, distribution-level feature–prototype assignment** to balance coverage and suppress noise for prompt generation rather than labels or features for downstream training.
> - The prompt construction step bridges distribution-level alignment and instance-level localization.
>
> Taken together, SPROUT demonstrates that a carefully composed inference-time pipeline can enable competitive dense instance segmentation without any supervised training. We believe this provides a useful perspective for adapting promptable foundation models to new domains without annotations.
>
> > W2&Q2: How robust is the proposed prototype construction based on optical density maps when the H&E color prior is corrupted?
>
> A: Thank you for the comment! We agree that severe staining corruption can weaken the reliability of the initial H&E prior. However, the Otsu-derived masks are used only for high-confidence biological initialization in prototype construction, not final segmentation cues. The subsequent similarity mapping and POT-scan stages can partially absorb imperfections by propagating prototype semantics at the distribution level and leaving the ambiguous mass unmatched. Empirically, under moderate appearance shifts induced by stain normalization, color jitter, and OD scaling, we observe only modest degradation, suggesting that the framework does not depend heavily on exact threshold quality as long as the prior still captures a reasonably clean subset of nuclear and tissue cues.
>
> At the same time, under severe image corruption, initialization can become less reliable. This remains a limitation of SPROUT and is shared by pathology image segmentation methods. We hope this addresses your concerns, and we welcome any further discussion.
>
> | Setting | AJI | Dice |
> | :--- | :---: | :---: |
> | Macenko | 0.619 | 0.799 |
> | Vahadane | 0.614 | 0.793 |
> | BCS ± 0.1 | 0.617 | 0.792 |
> | BCS ± 0.2 | 0.611 | 0.790 |
> | H ± 0.02 | 0.616 | 0.791 |
> | H ± 0.05 | 0.612 | 0.789 |
> | BCS ± 0.1, H ± 0.02 | 0.618 | 0.792 |
> | BCS ± 0.2, H ± 0.05 | 0.612 | 0.790 |
> | H×0.8 | 0.604 | 0.782 |
> | E×0.8 | 0.607 | 0.783 |
> | H×1.2 | 0.608 | 0.784 |
> | E×1.2 | 0.609 | 0.785 |
> | H×0.8, E×1.2 | 0.602 | 0.784 |
> | H×1.2, E×0.8 | 0.606 | 0.783 |
> | SPROUT | 0.621 | 0.795 |
>
> > Q3: Why is POT used to identify positive prompts, while a simple uniform sampling is used for negative prompts?
> >
>
> A: Thank you for the observation! The different sampling strategies reflect the distinct **roles** of positive and negative prompts in SAM.
>
> **Positive prompts** must identify the target nucleus precisely. A misplaced positive point can directly shift the prediction to the wrong instance. As discussed in Section F, SAM generally benefits from positive points located in the more representative interior rather than near ambiguous boundaries. Therefore, we use POT-based foreground assignment to obtain reliable positive seeds.
>
> **Negative prompts**, in contrast, serve as exclusion cues to prevent mask expansion into the surrounding background or neighboring nuclei. For this role, broad spatial coverage is more important than exact semantic assignment. We therefore adopt a simple **grid-based uniform sampling** strategy that imposes a grid on the background mask and randomly samples points from valid grid cells. Compared with standard grid sampling, this **low-cost** design provides **broader background coverage and better captures narrow inter-cell gaps**. We will include this clarification in the revised version.

---

> > ### Author Rebuttal · Reviewer_15wc · 2026-04-04
> >
> > Thank you for the rebuttal. We have udated the score accordingly.

---

> > > ### Author Response · Authors · 2026-04-04
> > >
> > > Dear Reviewer `15wc`,
> > >
> > > Thank you very much for your thoughtful follow-up and for reading our rebuttal. We are truly grateful to know that our revisions and clarifications helped address your concerns.
> > >
> > > We also sincerely appreciate your thoughtful reassessment of the paper. Thank you again for your valuable feedback and for the effort you put into the review process.
> > >
> > > Best regards,
> > >
> > > The Authors of Submission 19778

---

### Official Review · Reviewer_RheG · 2026-03-13

**Soundness:** 3
**Presentation:** 3
**Significance:** 3
**Originality:** 2
**Overall Recommendation:** 4
**Confidence:** 4

**Summary:**

This paper proposes SPROUT, a training- and annotation-free prompting framework for nuclear instance segmentation in histopathology. The method leverages histology-informed priors to construct slide-specific reference prototypes, which progressively guide feature alignment via a partial optimal transport scheme. The aligned foreground and background features are then converted into positive and negative point prompts to drive the SAM without any parameter updates. Experiments across multiple histopathology benchmarks demonstrate competitive performance compared to supervised and fine-tuned approaches.

**Compliance With Llm Reviewing Policy:**

Affirmed.

**Final Justification:**

My most concerns have been addressed.

**Key Questions For Authors:**

see the above Weaknesses

**Limitations:**

yes

**Strengths And Weaknesses:**

Strengths
1. The idea of performing nuclear instance segmentation without any training or annotation is interesting.
2. The paper includes comparisons against multiple baselines and provides reasonably ablation studies to analyze the contributions of different components.
3. The manuscript is clearly written and easy to follow. The methodological pipeline and experimental setup are described in a structured and understandable manner.

Weaknesses
1. Unclear motivation and limited technical novelty: The motivation for pursuing a fully training-free nuclear instance segmentation approach is not entirely convincing. If training or lightweight fine-tuning can significantly improve performance, such improvements may have greater clinical relevance than a strictly training-free solution. The paper does not sufficiently justify why eliminating training is inherently preferable in practical clinical workflows.
Furthermore, the technical novelty appears limited. Prototype learning and optimal transport-based alignment have been extensively studied in segmentation literature (e.g., [1–4]). While SPROUT integrates these ideas within a prompting framework, the paper does not clearly articulate how this integration constitutes a substantial methodological advancement. A more detailed discussion of related work and clearer differentiation from prior prototype-based and OT-based segmentation methods are necessary.

2. Inference efficiency analysis is incomplete: Figure 6 analyzes the relationship between different SAM variants, AJI performance, and inference time. However, the paper does not compare the inference time of SPROUT against other nuclear instance segmentation methods listed in Table 1. Although the method is described as “training-free,” the inference pipeline still involves multiple non-trivial steps (prototype construction, clustering, optimal transport alignment, prompt generation, and SAM inference). A direct comparison of inference time with existing nuclear instance segmentation approaches would provide a more complete picture of its practical efficiency.

3. Strong dependence on SAM and clustering quality: The performance of SPROUT appears highly dependent on the underlying SAM backbone and the quality of clustering/prototype construction.
* If SAM is replaced with a domain-adapted variant such as MedSAM, how would performance change?
* How sensitive is the method to different clustering strategies?
* Does the proposed alignment mechanism still provide gains when using stronger backbones or alternative clustering methods?

---

> ### Author Rebuttal · Authors · 2026-03-31
>
> Dear Reviewer `RheG`,
>
> Thank you for your thoughtful feedback! We have carefully revised the manuscript to address each point raised.
>
> ---
>
> >W1:  Why is a strictly training-free design useful in practical clinical workflows, and what is SPROUT’s main novelty?
>
> A: Thanks for the comment! Our goal is to show that a designed prompting pipeline can unlock strong nuclear instance segmentation from SAM without parameter updates, reducing deployment friction. It  is often preferable in clinical workflows where
>
> - pixel-level annotations are costly and time-consuming,
> - data are privacy-restricted and institution-specific,
> - distribution shifts across staining/scanners require repeated retraining.
>
> Practically, training-free methods are attractive in multi-center deployment, where repeated annotation and retraining can quickly outweigh the marginal benefit of full supervision. In such settings, training-free deployment is not just convenient, but the more viable choice.
>
> Regarding novelty, while prototype learning and OT are established, SPROUT makes three key advances:
>
> - Conventional methods rely on task-specific architectures, SAM adaptation, and high-quality annotations. SPROUT enables **frozen** SAM to achieve competitive performance through inference-time prompting in a training-free setting.
> - Prior prototypes and OT-based methods focus on semantic segmentation, few-shot matching, or natural-image settings with limited objects. SPROUT targets **large-scale** instance segmentation without external references.
> - SPROUT incorporates **biological priors** rather than relying solely on generic visual similarity to guide segmentation, improving both task relevance and interpretability.
>
> >W2: How efficient is SPROUT in practice?
>
> A: Thank you for the comment! We have added both the stage-wise breakdown and a direct comparison on CPM17. SPROUT is slower at test time than a lightweight feed-forward model such as U-Net, with the main bottlenecks arising from POT-scan and SAM prediction. Nevertheless, the trade-off is **higher test-time latency in exchange for lower target-dataset setup cost**. Compared with MedSAM, SPROUT has moderately higher inference time with better performance, while COIN is faster at test time but has the highest setup cost due to SAM-involved training.
>
> | Stage | Init. mask  | Proto. construction | POT | Prompting | SAM | Post-proc. |
> | :--- | :---: | :---: | :---: | :---: | --- | :---: |
> | Time (s)  | 0.071 | 0.963 | 1.991 | 0.027 | 5.174 | 1.232 |
>
> | Method | Train? | Ann.? | New data retrain? | s/img | AJI ↑ |
> | :--- | :---: | :---: | :---: | :---: | :---: |
> | U-Net | ✓ | ✓ | ✓ | 0.294 | 0.554 |
> | MedSAM | ✓ | ✓ | ✗ | 7.242 | 0.648 |
> | COIN | ✓ | ✗ | ✓ | 4.311 | 0.572 |
> | SPROUT | ✗ | ✗ | ✗ | 9.452 | 0.662 |
>
> >Q1: If SAM is replaced with a domain-adapted variant such as MedSAM, how would performance change?
>
> A: Thank you for the question! We replace SAM with MedSAM and SAM-Med2D [1] with their publicly available checkpoints built on SAM-b. The results remain close to those of SPROUT w/ SAM-b, suggesting that our pipeline effectively unlocks strong segmentation capability even with the original backbone.
>
> |  | MoNuSeg |  | CPM17 |  |
> | :--- | :---: | :---: | :---: | :---: |
> |  | AJI | Dice | AJI | Dice |
> | w/ MedSAM | 0.609 | 0.786 | 0.654 | 0.811 |
> | w/ SAM-Med2D | 0.606 | 0.782 | 0.651 | 0.809 |
> | w/ SAM-b | 0.615 | 0.787 | 0.649 | 0.803 |
>
> [1] J. Cheng, et al. “SAM-Med2D”, arXiv 2308.16184.
>
> >W3&Q2: How sensitive is SPROUT to clustering?
>
> A: Thank you for the question! We evaluated K-means++ and GMM on MoNuSeg. The AJI varies within 0.01 across prototype numbers, indicating limited sensitivity to clustering choice. GMM is slightly worse, which is consistent with the intuition that soft assignments produce less distinctive prototype centers.
>
> |  | 2 | 3 | 4 | 5 | 6 |
> | :--- | :---: | :---: | :---: | :---: | :---: |
> | K-means | 0.614 | 0.621 | 0.618 | 0.616 | 0.620 |
> | K-means++ | 0.613 | 0.620 | 0.616 | 0.619 | 0.612 |
> | GMM | 0.610 | 0.613 | 0.607 | 0.606 | 0.609 |
>
> >W3&Q3: Does alignment still provide gains with stronger backbones or other clustering methods?
>
> A: Thank you for the question! We have added results from Phikon-v2 [2] and Prov-GigaPath [3], alongside pathology-specific and general-domain backbones (Section 4.1 and Section E.1). Across all backbones, the proposed alignment mechanism consistently maintains strong performance.
>
> As shown in Q2, replacing clustering methods causes only marginal changes, suggesting that the gain is robust to clustering strategies.
>
> |  | MoNuSeg |  | CPM17 |  |
> | :--- | :---: | :---: | :---: | :---: |
> |  | AJI | Dice | AJI | Dice |
> | Phikon-v2 | 0.613 | 0.785 | 0.655 | 0.817 |
> | Prov-GigaPath | 0.618 | 0.789 | 0.650 | 0.811 |
>
> [2] A. Filiot, et al. “Phikon-v2, A large and public feature extractor for biomarker prediction”, arXiv 2409.09173.
>
> [3] H. Xu, et al. “A whole-slide foundation model for digital pathology from real-world data”, Nature 2024.

---

> > ### Author Rebuttal · Reviewer_RheG · 2026-04-03
> >
> > Thanks for the detailed rebuttal, which has addressed most of my concerns.

---

> > > ### Author Response · Authors · 2026-04-03
> > >
> > > Dear Reviewer `RheG`,
> > >
> > > We truly appreciate your time and consideration in reviewing our manuscript and rebuttal. Your valuable comments helped us sharpen the positioning of our contribution and strengthen the efficiency and robustness.
> > >
> > > It is very encouraging to know that our responses have fully addressed your main concerns. If you feel that our rebuttal now better clarifies the contribution and technical soundness of the work, we would be sincerely grateful if you would also consider adjusting the score. Thank you again for your thoughtful assessment.
> > >
> > > Best regards,
> > >
> > > The Authors of Submission 19778

---

### Decision · Program_Chairs · 2026-04-30

**Decision:**

Accept (regular)

**Comment:**

This paper introduces a pipeline for segmenting individual nuclei in histopathology images that requires no training or manual annotation. It sequentially performs stain deconvolution to identify foreground and background regions, then produces foreground and background prototypes which are aligned to whole-image features via an optimal transport process. The result is used to produce positive and negative point prompts for SAM, which are then post-processed to minimize errors. Evaluation on several histopathology benchmarks indicates strong performance in comparison to supervised and fine-tuned alternatives. The reviewers agree that the proposed approach is sound and well motivated, and that the experimental results are convincing. While the pipeline involves substantial engineering and the application is relatively narrow, the work is a proof of concept for prompt design to leverage foundational models and its application in practice, which could have significant impact.